# Post-translational modification-dependent oligomerization switch in regulation of global transcription and DNA damage repair during genotoxic stress

Prathama Talukdar[1], Sujay Pal[1,2] & Debabrata Biswas [1] ✉

Mechanisms of functional cross-talk between global transcriptional repression and efficient DNA damage repair during genotoxic stress are poorly known. In this study, using human AF9 as representative of Super Elongation Complex (SEC) components, we delineate detailed mechanisms of these processes. Mechanistically, we describe that Poly-Serine domain-mediated oligomerization is pre-requisite for AF9 YEATS domain-mediated TFIID interaction-dependent SEC recruitment at the promoter-proximal region for release of paused RNA polymerase II. Interestingly, during genotoxic stress, CaMKII-mediated phosphorylation-dependent nuclear export of AF9-specific deacetylase HDAC5 enhances concomitant PCAF-mediated acetylation of K339 residue. This causes monomerization of AF9 and reduces TFIID interaction for transcriptional downregulation. Furthermore, the K339 acetylation-dependent enhanced AF9-DNA-PKc interaction leads to phosphorylation at S395 residue which reduces AF9-SEC interaction resulting in transcriptional downregulation and efficient repair of DNA damage. After repair, nuclear re-entry of HDAC5 reduces AF9 acetylation and restores its TFIID and SEC interaction to restart transcription.

In response to exposure to DNA damage-inducing genotoxic stress, mammalian cells employ initial global transcription inhibition for avoiding fatal collision between ongoing transcription and DNA repair apparatus leading to genomic instability, as well as for gaining proper access to machineries involved in efficient repair of the damaged DNA[1,2]. Ubiquitin-mediated RNA polymerase II (Pol II, hereafter) degradation, which is the "last-resort mechanism", is one of the widely accepted mechanisms for global transcription inhibition[3,4]. However, depending on the extent of DNA damage, especially when challenged with low dosage of DNA-damaging agents from external sources, the cells employ transient regulation of transcription as well as DNA repair for efficient maintenance of overall processes. In this context, two recent studies have shown a role in identification of DNA lesion

through ongoing transcription machineries, which is critical for overall efficient repair of damaged DNA and maintenance of genomic integrity[5,6]. Moreover, since after release of transcriptionally-engaged Pol II from promoter region, elongation factors regulate overall downstream events across the coding region, hence it is conceivable that these factors, along with associated regulators, could play key roles in transcription regulation as well as DNA repair upon exposure to genotoxic stress.

Among the elongation factors, human super elongation complex (SEC) comprising AFF1/AFF4, ELL, AF9/ENL, EAF1/EAF2, and P-TEFb complex, plays key roles in transcription elongation by Pol II of large number of genes[7–9]. With the exception of EAF1/2 and P-TEFb (a heterodimer of CDK9 and CyclinT1/T2), all other SEC components

[1]CSIR-Indian Institute of Chemical Biology, Kolkata 700032, India. [2]Academy of Scientific and Innovative Research (AcSIR), Ghaziabad 201002, India. ✉ e-mail: dbiswas@iicb.res.in

frequently fuse with N-terminus of Mixed Lineage Leukemia (MLL) to form MLL fusion proteins that, in conjunction with wild type MLL, gives rise to pediatric acute form of both myeloid and lymphoid leukemia through aberrant transcriptional misregulation of *HOX* cluster genes during hematopoiesis[7]. Emphasizing a role for these elongation factors in temporal regulation of transcription, two of our recent studies have deciphered a role of p300-mediated acetylation of AFF1 and ATM-mediated phosphorylation of ELL in global transcription inhibition upon exposure to genotoxic stress[10,11]. Further, ATM-mediated phosphorylation of ENL has also been shown to be important for its interaction with PRC1 complex and its subsequent recruitment at the DSB site for transcription repression[12]. However, understanding of the overall integrated mechanisms involving these post-translational modifications for their transcriptional regulation, and mechanisms of functional recovery after repair of damaged DNA for transcription restart, remains poorly understood. In this study, using AF9 as a model elongation factor, we decipher the detailed mechanisms of these integrated events.

Human AF9, an AF9-family protein (containing ENL homolog), harbors an N-terminal YEATS domain for recognition of acetylated and crotonylated histone H3[13–15] for its recruitment on target genes and for interaction with PAF1 complex (PAF1c) and DNA[16,17]. The C-terminal SEC-interacting domain of AF9 frequently fuses with MLL giving rise to MLL-AF9 fusion protein[7]. Besides these domains, AF9 also contains a stretch of Serine residues (Poly-Ser, hereafter) that we have shown to play crucial roles in AF9-mediated transcriptional activation through interaction with TFIID complex for SEC recruitment at the promoter-proximal region for efficient pause-release of Pol II[18,19]. Further, continuing with detailed mechanistic understanding of AF9 in regulation of transcription activation, in this study, we have identified a role of YEATS domain of AF9 in its interaction with TFIID, that is critically dependent on the Poly-Ser domain-mediated oligomerization of AF9. This oligomerization-dependent transcriptional activation by AF9, is being elegantly used as switching mechanism during genotoxic stress for temporal regulation of global transcription and DNA repair as well as restart afterwards that involves post-translational modifications of key residues within AF9. Thus, our study has deciphered evolutionary-conserved elegant mechanisms that mammalian cells have evolved for responding to transient exposure to genotoxic stress for optimal transcriptional regulation and maintenance of genomic integrity and cell survival.

## Results

### Both Poly-Ser and YEATS domains are indispensable for AF9 interaction with TFIID, but not SEC

As shown in Supplementary Fig. 1A, human AF9 contains C-terminal SEC-interacting and N-terminal YEATS domain for interaction with histone acetylation and crotonylation modifications. Apart from these domains, we have recently reported an exclusive role for Poly-Ser domain in interaction with TFIID components (Supplementary Fig. 1B) for recruitment of SEC at the promoter-proximal region for release of paused Pol II into productive elongation[18,19]. For further understanding of the associated mechanism(s) and to address potential role of adjacent regions in the overall regulation, our serial deletion of 20 amino acids (aa) from the N-terminal end showed importance of AF9 residues (41–60 aa) in its interaction with TFIID components within mammalian cells by immunoprecipitation analyses (Fig. 1A, compare lane 4 vs lane 5). Further deletion from 81–100 amino acids almost abrogated this interaction (lane 7), whereas, deletion up to 112 amino acids, that comprise the YEATS domain, completely abolished AF9 interaction with TFIID (lane 8). These observations suggest that the deletion fragment that retains the Poly-Ser domain (149–196 aa) fails and is not sufficient on its own to interact with TFIID. A similar analysis with internal deletion of Poly-Ser as well as YEATS domain within the full-length AF9 protein (Fig. 1B) showed the importance of both these

domains of AF9 protein in its interaction with TFIID, without having any effect on interaction with SEC components such as ELL and CDK9 (Fig. 1B, compare lane 2 vs lanes 3, 4). Thus, our interaction analyses uncovered an unexpected role of YEATS domain that, in conjunction with Poly-Ser domain, is important for interaction with TFIID.

Next, for identifying the specific region within YEATS domain that would be important for AF9-TFIID interaction, we generated a small internal deletion construct within the 41–60 amino acids since a deletion of this region markedly abrogated AF9 interaction with TFIID in our deletion analyses (Fig. 1A, lane 5). Within this region, sequence conservation was observed between 43–60 amino acids across multiple YEATS domain-containing proteins (Supplementary Fig. 1C) suggesting the importance of these sequences in overall functional regulation. Interestingly, an internal deletion of this short stretch of amino acids also impaired AF9 interaction with TFIID without showing any effect on SEC interaction (Fig. 1C, compare lane 2 vs lane 4). Remarkably, this region forms a distinct surface than the one involved in recognition of acetyl lysine motif through formation of interacting pocket as defined by an earlier study (Supplementary Fig. 1D)[13]. Therefore, it seems that the human AF9 uses multiple surfaces in its interaction with several interacting proteins for functional regulation of transcription within mammalian cells.

### YEATS domain-mediated TFIID interaction is important for expression of diverse AF9-target genes

For addressing the importance of YEATS domain-mediated AF9-TFIID interaction in transcriptional regulation, we initially generated stable AF9 knockdown (KD) cells by employing several shRNAs as shown in Fig. 1D. Our RNA analyses by qRT-PCR using one of these KD cells, showed reduced expression of several AF9-target genes as reported in our earlier RNA-Seq experiment (Fig. 1E)[18]. The overall effect is specific since in the same assay, we have failed to observe any effect of knockdown on expression of non-target genes such as *CDKN2B*, *KLF5*. Subsequent chromatin immunoprecipitation (ChIP) analyses in AF9 KD cells showed significantly reduced recruitment of SEC components ELL and P-TEFb complex (using CDK9 as representative) that resulted in reduced presence of phosphorylated form of Pol II CTD at Ser2 and Ser5 residues leading to elevated presence of transcriptionally-engaged paused Pol II at the promoter-proximal region of some of these target genes that we have tested (Fig. 1F). Consistent with a role of enhanced pausing in reducing recruitment of upstream transcription initiation factors[18,20], we have also observed reduced recruitment of TFIID (TBP as representative) at the promoter-proximal region upon AF9 KD. The overall effect of KD on factor recruitment at the target genes is specific since we have observed a dissimilar, yet opposite, trend of factor binding on non-target gene such as *CDKN2B*(Fig. 1F). Thus, based on the RNA expression and ChIP analyses, several genes have been identified which would be subsequently used as representative of AF9-target genes for detailed mechanistic understanding of relevance of regulation of bi-functional AF9 protein in transcription processes and its effect on gene expression within mammalian cells in a context-dependent manner.

Once identified, to address the role of YEATS domain-mediated TFIID interaction in regulation of AF9-target gene expression, we re-expressed wild type (WT) as well as YEATS domain mutant (43–60 aa Δ) in AF9 KD cells through their ectopic expression, close to the endogenous level for avoiding the artifacts associated with over-expression of proteins (Fig. 1G, compare lane 1 vs lanes 4, 5). Interestingly, re-expression of AF9 wild type ((AF9 (WT)) significantly restored the expression of AF9-target genes close to the control scramble (Scr) cells, whereas, the parallel analyses with control empty vector (EV) failed to do so (Fig. 1H). Interestingly, the YEATS domain mutant (43–60 aaΔ) that showed defective TFIID interaction despite containing the Poly-Ser domain (Fig. 1C), failed to restore expression of target genes in AF9 KD cells when compared to the WT control

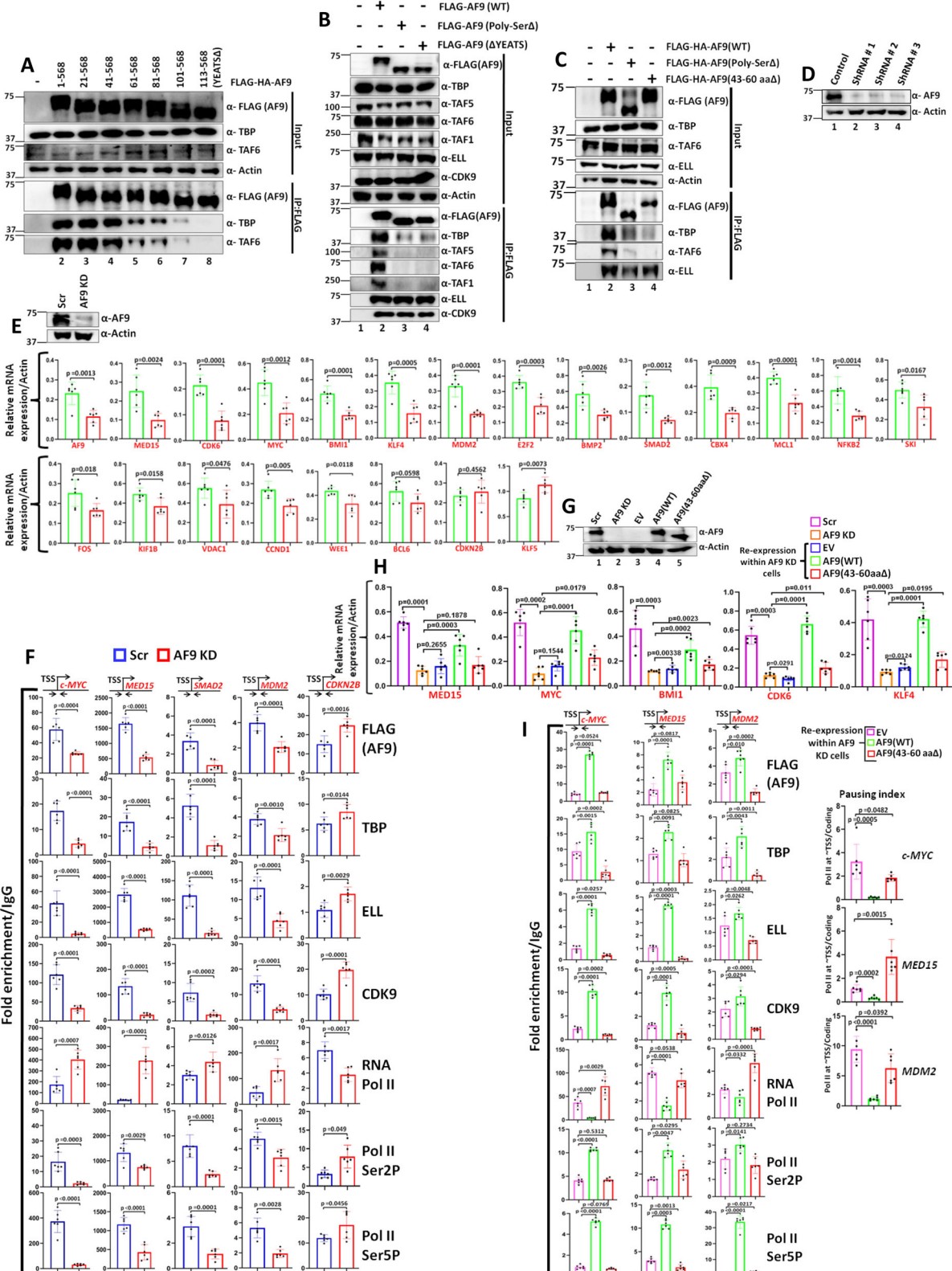

(Fig. 1H). These results, thus suggest that a functional interaction between AF9 YEATS domain and TFIID is important for target gene expression within mammalian cells.

Subsequently, we performed ChIP analyses in these cells (Fig. 1H) for addressing requirement of YEATS domain-mediated TFIID interaction in target gene expression. AF9 KD cells transfected with control EV showed significantly low level of AF9 and majority of other factors

recruitment (Fig. 1I). As compared to EV, AF9 KD cells transfected with AF9(WT) showed restoration of recruitment of AF9 coupled with all the factors including SEC components ELL and CDK9 as well as TFIID. This resulted in enhanced level of Pol II CTD Ser2 and Ser5 phosphorylation that led to release of paused Pol II being present at the promoter-proximal region (Fig. 1I, compare EV vs AF9 WT). More importantly, re-expression of AF9 YEATS domain mutant (43−60 aaΔ),

**Fig. 1 | Besides Poly-Ser, AF9 also requires YEATS domain for TFIID interaction and transcriptional activation within 293 T cells. A** Immunoblotting analysis showing interaction of ectopically expressed N-terminal-deleted AF9 fragments with TFIID components as indicated ($n = 3$ replicates). **B** Immunoblotting analysis showing the effect of AF9 YEATS and Poly-Ser domain deletion on its interaction with TFIID and SEC components as indicated ($n = 3$ replicates). **C** Immunoblotting analysis showing the requirement of N-terminal minimal region (43–60 aa) of AF9 for its exclusive interaction with TFIID ($n = 2$ replicates). **D** Immunoblotting analysis showing stable knockdown of endogenous AF9 protein by multiple shRNAs using actin as a loading control ($n = 3$ replicates). **E** qRT-PCR analysis showing effect of AF9 stable knockdown (KD) on mRNA expression of indicated AF9-target genes. The inset immunoblot panel shows a knockdown of AF9 in the cells that were used for the RNA analysis ($n = 2$ replicates). **F** ChIP analysis showing the effect of AF9 stable knockdown on recruitment of indicated target factors at the promoter-proximal region of selected AF9-target genes (from (**E**)) as mentioned ($n = 2$

replicates). **G** Immunoblotting analysis showing re-expression of AF9(WT) and AF9(43–60 aaΔ) proteins in AF9 KD cells through their ectopic expression ($n = 2$ replicates). **H** qRT-PCR analysis showing effect of re-expression of AF9(WT) and AF9(43–60 aaΔ) proteins in stable AF9 KD cells on mRNA expression of indicated AF9-target genes ($n = 2$ replicates). **I** ChIP analysis showing effect of re-expression of AF9(WT) and AF9(43–60 aaΔ) proteins in stable AF9 KD cells on recruitment of indicated target factors at the promoter-proximal region of selected AF9-target genes (left panel). Parallel use of empty vector (EV) was used as a control in our experimental setup. Statistical significance was calculated as Scr vs AF9 KD, and AF9 KD vs EV, AF9(WT), AF9(43–60 aaΔ). Pausing index analysis of Pol II on the selected AF9-target genes as indicated (right panel) ($n = 2$ replicates). In this figure, for data showing RNA and ChIP analyses by qRT-PCR, the error bar represents mean ± SD and statistical analyses were performed using one-tailed Student's $t$ test. $p$ values for each experimental data is indicated on the bar diagram.

that failed to interact with TFIID, despite retaining full SEC interaction capability, failed to get recruited at the promoter-proximal regions of target genes (Fig. 1I, compare WT vs 43–60 aaΔ). Consistent with its reduced TFIID-dependent recruitment, we also observed reduced SEC as well as TFIID recruitment similar to that observed for AF9 KD cells (Fig. 1I, compare EV vs 43–60 aaΔ), accompanied by reduced level of Pol II Ser2 and Ser5 phosphorylation leading to enhanced presence of paused Pol II at the promoter-proximal region of the target genes. Subsequent pausing index analysis (a ratio of level of Pol II at ~TSS/coding region) showed that while re-expression of AF9(WT) reduced pausing in the knockdown cells, similar expression of AF9 (43–60 aaΔ) failed to reduce the enhanced pausing observed in the knockdown cells (Fig. 1I, right panel, compare EV vs AF9(WT) and EV vs AF9 (43–60 aaΔ). Further, in accordance with a role of majority of AF9-target genes (Fig. 1E) in regulation of cell division and cell proliferation of mammalian cells, we have observed restoration of defective proliferation and colony formation ability observed in AF9 KD cells upon re-expression of AF9(WT) (Supplementary Fig. 1E, F). However, similar re-expression of the AF9 YEATS domain mutant (43–60 aaΔ) failed to do so. Overall, based on these data, we conclude that along with Poly-Ser domain, we have deciphered an additional role of YEATS domain of AF9 in its interaction with TFIID that is important for AF9-dependent SEC component recruitment at the promoter-proximal region for release of paused Pol II for optimal expression of target genes that are important for proliferation and colony formation ability of mammalian cells.

## Poly-Ser domain of AF9 is important for self-association and oligomerization of AF9 within mammalian cells

Based on our data as presented in Fig. 1 as well as our earlier reported study[18], it is evident that both the Poly-Ser as well as YEATS domain of AF9 are important for its interaction with TFIID for regulation of target gene expression. Human AF9 YEATS domain is a structured domain[13], whereas, the Poly-Ser domain does not predict to form any structure by AlphaFold structural prediction of entire AF9 protein (Supplementary Fig. 2A)[21,22]. It is interesting to note that although the YEATS domain and the SEC-interacting domain form closely juxtaposed structured domain surface, effect of Poly-Ser domain is observed only for YEATS domain-mediated interaction with the TFIID and not the SEC components. What is the functional significance of Poly-Ser domain in this overall interaction of AF9 with TFIID, but not SEC components? Protein self-association has been shown to play important roles in differential association with cognate interacting proteins within mammalian cells[10,23]. For example, our earlier study has shown preferential association of dimeric and monomeric forms of ZMYND8 with the transcriptional activator PTEF-b and the repressor NuRD complex, respectively[23]. Keeping this in mind, we initially addressed whether AF9 protein would also show self-association property within mammalian cells. As shown in Fig. 2A, ectopically expressed AF9 proteins

showed self-association when they are co-expressed within mammalian cells (lane 4). Consistent with this observation, immuno-fluorescence study also showed strong colocalization of co-expressed GFP-AF9 and FLAG-AF9 (Supplementary Fig. 2B). Further, to show a direct evidence of this self-association, bacterially-expressed and purified GST-AF9 and His-GFP-AF9 (Supplementary Fig. 2C, D) also showed interactions among AF9 proteins in vitro (Supplementary Fig. 2E, lane 5). Overall, these data suggest that the human AF9 protein shows self-association both in vitro and in vivo within mammalian cells.

Next, using several deletion constructs, we addressed the requirement of specific domain within AF9 for their self-association. As shown in Fig. 2B, AF9 fragments that contain the Poly-Ser domain (lanes 4–8), fully retained self-association property. However, fragment without the Poly-Ser domain (lane 9, 270–568) lost the ability to self-associate. Subsequent immunoprecipitation assay using internal deletion of Poly-Ser domain further confirmed the requirement of this domain for self-association of AF9 (Fig. 2C, compare lane 4 vs lane 5) without the requirement of YEATS domain (Supplementary Fig. 2F, compare lane 4 vs lane 6). Therefore, based on all these evidences, we conclude that the Poly-Ser domain of AF9 is important for its self-association.

The self-associated proteins tend to oligomerize for their functional regulation. Consistent with this hypothesis, ectopically expressed FLAG-AF9 readily showed oligomerization upon mild cross-linking in presence of 0.02% glutaraldehyde, whereas, the Poly-Ser domain-deleted AF9 protein failed to show any oligomerization (Fig. 2D, compare lane 3 vs lane 4). Likewise, endogenous AF9 also showed oligomerization property (Fig. 2E). Based on the mobility, it appeared that AF9 forms predominantly trimer upon its oligomerization (Fig. 2D, E). However, endogenous AF9 also appeared to form higher oligomers as well. Similar to our observations within mammalian cells, the purified recombinant AF9 (Supplementary Fig. 2D) also showed oligomerization in native polyacrylamide gel and presence of predominantly trimer and monomer species (Fig. 2F). Thus, based on all these evidences, we conclude that owing to its Poly-Ser domain-dependent self-association property, human AF9 protein oligomerizes and predominantly forms trimer upon oligomerization.

## Poly-Ser domain-dependent oligomerization is critical for YEATS domain-mediated AF9 interaction with TFIID

For addressing the Poly-Ser domain-mediated oligomerization in regulation of TFIID interaction, we have generated two AF9 proteins containing 22 and 36 Serine residues deleted within the Poly-Ser amino acids (Supplementary Fig. 1A). Interestingly, deletion of 22 Serine residues modestly affected its self-association ((Supplementary Fig. 2G, compare lane 4 vs lane 5)) as well as oligomerization (Supplementary Fig. 2H, compare lane 5 vs lane 6). However, deletion of 36 Serine residues further reduced its self-association (Supplementary Fig. 2G, compare lane 4 vs lane 6), while it showed almost complete

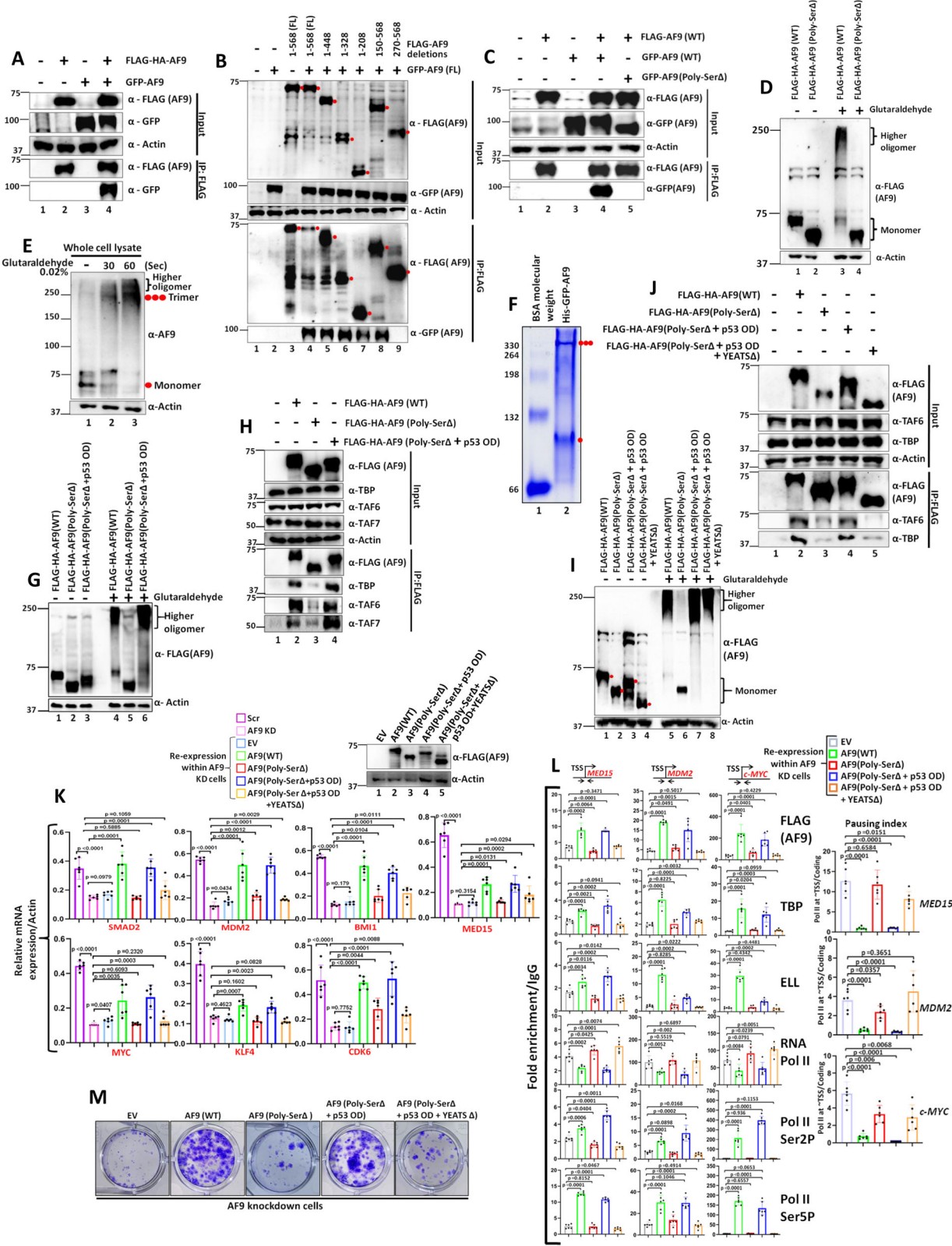

abolition of oligomerization like that of Poly-Ser-deleted constructs (Supplementary Fig. 2H, compare lane 5 vs lanes 7, 8). Deletion of Poly-Ser domain did not affect subcellular localization of AF9 proteins that showed its presence only within nucleus of mammalian cells like that of wild type (Supplementary Fig. 2I, J). These results thus rule out an effect of changes in subcellular localization of AF9 protein, upon deletion of Poly-Ser domain, in regulation of oligomerization capacity

of AF9. Since Poly-Ser-deleted AF9 shows impaired TFIID interaction (Fig. 1B,C), we hypothesized that this domain-mediated oligomerization would be critical for YEATS domain-mediated TFIID interaction. Similar to the degree of effect on oligomerization, we have also observed varying effect on TFIID interaction of AF9 upon deletion of Serine residues, in which, the 22 Serine residues deletion showed moderate effect on reducing TFIID interaction, whereas, the 36 Serine

**Fig. 2 | Poly-Ser domain-mediated oligomerization of AF9 protein is essential for its interaction with TFIID for transcriptional activation of target genes within 293 T cells. A–C** Immunoblotting analysis showing self-association between ectopically expressed FLAG-HA-AF9 and GFP-AF9 (**A**), $n = 4$ replicates), different AF9 domains (**B**), $n = 2$ replicates), and between ectopically expressed FLAG-HA-AF9(WT) and GFP-AF9 (WT and Poly-SerΔ) (**C**), $n = 3$ replicates) within mammalian cells. **D** Immunoblotting assay showing oligomerization of ectopically expressed FLAG-HA-AF9(WT) and FLAG-HA-AF9(Poly-SerΔ) proteins ($n = 2$ replicates). **E** Immunoblotting analysis showing oligomerization of endogenous AF9 protein with indicated period of incubation time ($n = 3$ replicates). **F** Native PAGE Coomassie staining showing formation of trimer and monomer by purified recombinant His-GFP-AF9 protein. Filled red single dot indicates monomer, and triple dot indicates trimer formation ($n = 3$ replicates). **G** Immunoblotting assay showing restoration of oligomerization capacity of AF9(Poly-SerΔ) protein upon introduction of oligomerization domain (OD) of p53 protein ($n = 2$ replicates). **H** Immunoblotting analysis showing restoration of interaction of TFIID components

with AF9(Poly-SerΔ) protein upon addition of p53 OD within this fragment ((AF9(Poly-SerΔ + p53 OD)) ($n = 2$ replicates). **I** Immunoblotting assay using cell lysates containing indicated AF9 proteins showing YEATS domain-independent oligomerization of AF9 ($n = 2$ replicates). **J** Immunoblotting analysis showing absolute requirement of YEATS domain for AF9 interaction with TFIID components ($n = 1$ replicate). **K** qRT-PCR analysis showing effect of re-expression of AF9(WT), AF9(Poly-SerΔ), AF9(Poly-SerΔ + p53 OD) and AF9(Poly-SerΔ + p53 OD + YEATSΔ) proteins in stable AF9 KD cells on mRNA expression of indicated AF9-target genes ($n = 2$ replicates). **L** ChIP analysis showing effect of re-expression of indicated AF9 proteins in stable AF9 KD cells on recruitment of indicated target factors at the promoter-proximal region of indicated AF9-target genes ($n = 2$ replicates). **M** Colony formation assay showing restoration of colony formation ability of stable AF9 KD cells that re-express indicated AF9 proteins ($n = 2$ replicates). For this figure, the data showing RNA and ChIP analyses by qRT-PCR, the error bar represents mean ± SD and statistical analyses were performed using one-tailed Student's $t$ test. $p$ values for each experimental data is mentioned on the bar diagram.

residues deletion showed almost abolition of TFIID interaction as observed for Poly-Ser deletion (Supplementary Fig. 2K, compare lane 2 vs lanes 3, 4 in the IP panel). In all these assays, we have observed no effect on SEC interaction. Thus, the effect of Poly-Ser domain-mediated oligomerization is restricted to its interaction with TFIID only.

Next, for providing a proof of concept for sole function of Poly-Ser domain in oligomerization of AF9 for YEATS domain-mediated TFIID interaction, we created a construct in which the Poly-Ser domain was swapped with oligomerization domain (OD) of p53 which has been shown to form tetramer by multiple studies[24,25]. Interestingly, introduction of p53 OD in place of Poly-Ser domain restored the oligomerization property of AF9 (Fig. 2G, compare lane 5 vs lane 6), that also showed restoration of TFIID interaction (Fig. 2H, compare lane 3 vs lane 4). The YEATS domain does not participate in this oligomerization since further deletion of YEATS domain within the AF9 chimeric protein containing p53 OD in place of Poly-Ser domain, did not affect overall oligomerization within mammalian cells (Fig. 2I, compare lane 6 vs lanes 7, 8). However, this construct showed a loss of TFIID interaction despite retaining full oligomerization potential (Fig. 2J, compare lane 4 vs lane 5). The fact that similar functional interaction with TFIID can be achieved through swapping this domain with oligomerization domain of another protein, confirms the sole functional role of Poly-Ser domain is to aid in oligomerization of AF9 protein. Thus, we describe a functional role of AF9 YEATS domain in oligomerization-dependent interaction with TFIID components that may have functional regulation within mammalian cells.

### Oligomerization-dependent YEATS domain-mediated TFIID interaction is key for activation of target gene expression

For addressing overall role of oligomerization-dependent YEATS domain-mediated TFIID interaction in functional regulation of transcription, we re-expressed AF9(WT) as well as other chimeras in AF9 KD cells as indicated in Fig. 2K (top panel). Subsequent qRT-PCR analysis showed that while re-expression of AF9(WT) significantly restored expression of target mRNAs in AF9 KD cells when compared to EV control ((Fig. 2K, bottom panel, compare EV vs AF9 (WT)), the Poly-Ser-deleted AF9 fragment failed to do so for majority of the target genes when compared to WT (Fig. 2K, compare WT vs AF9 Poly-SerΔ). Interestingly, re-expression of the chimeric AF9 protein with Poly-SerΔ +p53 OD significantly restored target gene expression like that of WT, whereas, same chimera losing the YEATS domain failed to do so (Fig. 2K, compare WT vs AF9 Poly-SerΔ+p53 OD and AF9 Poly-SerΔ +p53 OD + YEATSΔ).

Subsequent ChIP analyses showed that, while the re-expression of AF9(WT) also restored AF9 recruitment and corresponding TFIID and SEC components leading to enhanced Pol II Ser2 and Ser5 phosphorylation for releasing paused Pol II, the AF9 Poly-SerΔ failed to do so

(Fig. 2L). Importantly, re-expression of the AF9 Poly-SerΔ+p53 OD chimera significantly restored all the factor recruitment including TFIID and SEC components. Restoration of these factors also resulted in enhanced Pol II CTD Ser2 and Ser5 phosphorylation that strongly correlated with release of paused Pol II. However, deletion of YEATS domain within the AF9 Poly-SerΔ+p53 OD chimera failed to restore recruitment of all these factors in AF9 knockdown cells that led to enhanced presence of paused Pol II (Fig. 2L). Subsequent pausing index analysis also showed that the AF9 constructs that restored TFIID interaction, also showed reduced pausing, while the mutants that failed to do so, could not reduce the pausing observed in the knockdown cells (Fig. 2L, right panel, compare EV vs other AF9 proteins as indicated). Thus, our studies until now have discovered a role of Poly-Ser domain-dependent oligomerization of AF9 in regulation of YEATS domain-mediated TFIID interaction. This interaction is important for AF9-dependent SEC component recruitment at promoter-proximal region for transcriptional activation of target genes. A failure of this recruitment results in increased pausing and inefficient transcriptional activation. These overall mechanisms are shown in Supplementary Fig. 2L. Further, consistent with a role of AF9-target genes in regulating cell growth and colony formation ability of cells, re-expression of the AF9 chimeric constructs that restored target gene expression, also restored the defective colony formation ability of AF9 KD cells (Fig. 2M).

### Exposure to genotoxic stress causes monomerization and concomitant reduction in TFIID as well as SEC interaction with AF9 that correlates with reduced transcription within mammalian cells

For addressing physiological importance of Poly-Ser domain-dependent oligomerization of AF9 and its effect on YEATS domain-mediated TFIID interaction and corresponding transcriptional regulation, we focused on genotoxic stress-dependent temporal regulation of global transcription within mammalian cells especially the initial global transcriptional repression[10,11,26,27]. We subjected 293 T cells to one-time exposure to ionizing radiation (IR), that predominantly causes double strand break, at a dose of 10 Gy and letting the treated cells recover afterwards. Interestingly, ectopically expressed AF9 showed dynamic oligomerization potential at different time points during recovery in which at early time points (1 hr and 2hrs), it showed reduced oligomerization and predominant monomerization, whereas, at later time point (8hrs), it regained its oligomerization and showed reduced monomerization (Supplementary Fig. 3A, compare lane 5 vs lanes 6–8). Similar results of reduced oligomerization were also observed for endogenous AF9 as well at 1 hr time point post IR treatment (Fig. 3A compare lane 3 vs lane 4). Further, along with reduced self-association and oligomerization, we also observed reduced association with TFIID

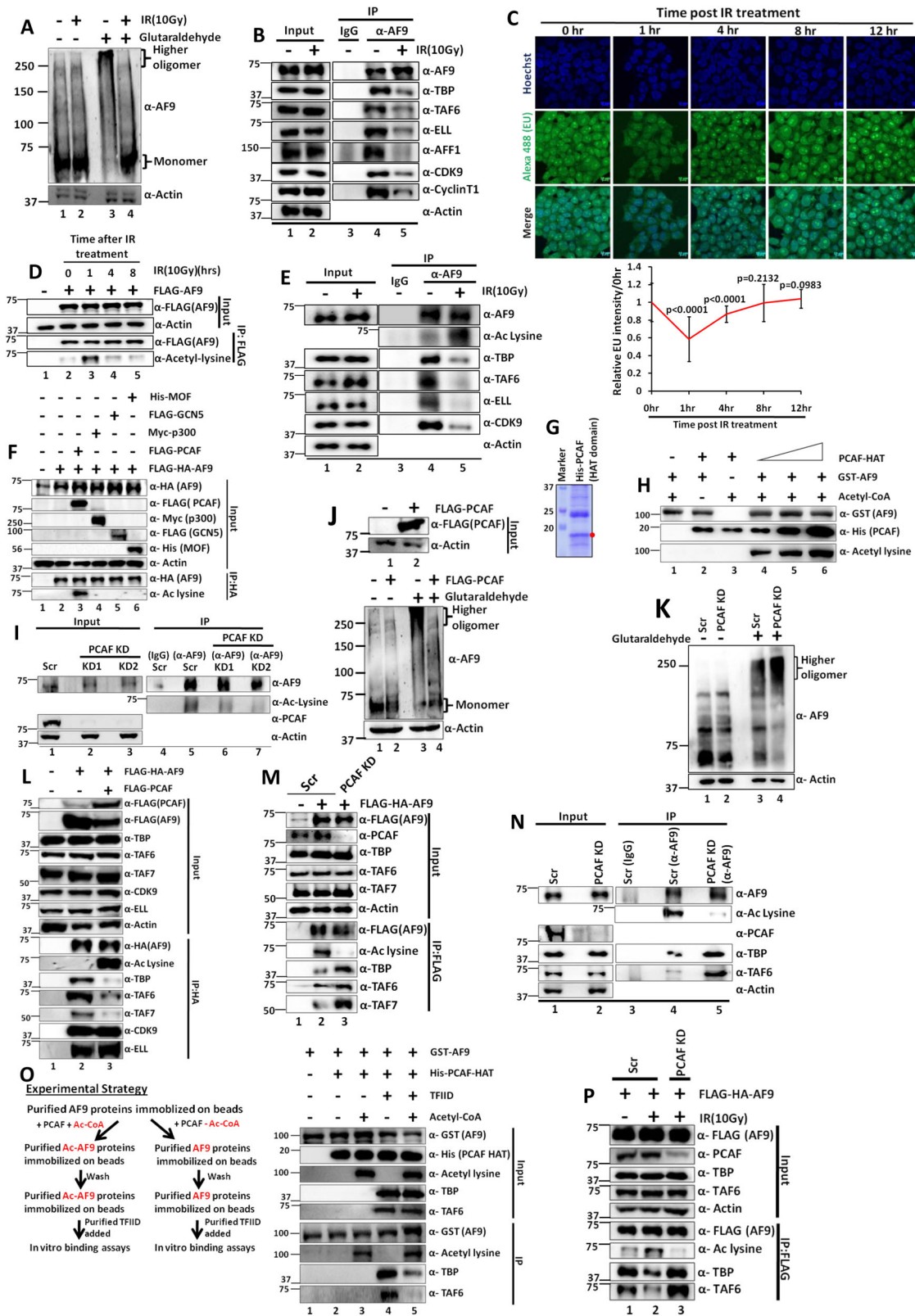

components such as TBP, TAF1, and TAF6 with ectopically expressed AF9 (Supplementary Fig. 3B, compare lane 2 vs lane 3) at early 1 hr time point post IR treatment. However, to our surprise, we also observed reduced association of SEC components ELL and CDK9 with ectopically expressed AF9 at 1 hr time point after IR treatment (Supplementary Fig. 3B, compare lane 2 vs lane 3). Similar results are also observed with endogenous AF9 as well that also showed reduced association

with both TFIID as well as SEC components at 1 hr time point post IR treatment (Fig. 3B, compare lane 4 vs lane 5).

To further understand whether reduction of this oligomerization and interaction would correlate with dynamic transcriptional activity within mammalian cells upon exposure to IR, we performed nascent RNA transcription analysis by labeling the RNA with uracil analog 5'-Ethynyl uridine (EU) at different time points after IR treatment[10,11].

**Fig. 3 | PCAF-mediated acetylation of AF9 reduces its oligomerization potential and TFIID interaction during exposure to genotoxic stress within 293 T cells.**
**A** Reduced oligomerization of endogenous AF9 within mammalian cells upon IR(10 Gy) treatment($n = 2$ replicates). **B** Immunoblotting analysis showing effect of IR(10 Gy) treatment on interaction of TFIID and SEC components with endogenous AF9 protein ($n = 3$ replicates). **C** Nascent RNA transcription analysis showing dynamic regulation of global transcription within mammalian cells upon IR(10 Gy) treatment. Error bar represents mean ± SD ($n = 100$ cells). This experiment was performed once. **D** Effect of IR-induced genotoxic stress on dynamic acetylation of ectopically expressed AF9 within mammalian cells ($n = 4$ replicates).
**E** Immunoblotting analysis showing effect of IR(10 Gy) treatment on acetylation of endogenous AF9 and its concomitant reduced interaction with TFIID and SEC components ($n = 3$ replicates). **F** Analysis of specificity of AF9 acetylation by various acetyl transferases as indicated ($n = 1$ replicate). **G** SDS Coomassie staining showing purification of recombinant HAT domain of PCAF ($n = 2$ replicates). **H** In vitro acetyl transferase assay showing efficient acetylation of purified GST-AF9 by PCAF HAT domain ($n = 1$ replicate). **I** Immunoblotting analysis showing effect of stable PCAF KD (using 2 different shRNAs) on acetylation of endogenous AF9 within mammalian cells ($n = 2$ replicates). **J** Immunoblotting analysis showing reduced oligomerization capacity of endogenous AF9 protein in presence of over-expressed PCAF ($n = 1$ replicate). **K** Effect of PCAF KD on oligomerization capacity of endogenous AF9 protein within mammalian cells ($n = 1$ replicate). **L** Effect of PCAF-mediated acetylation of ectopically expressed AF9 on its interaction with endogenous TFIID and SEC components as indicated; ($n = 3$ replicates). **M, N** Effect of stable knockdown of PCAF on acetylation of ectopically expressed AF9 ((**M**); n = 2 replicates)and endogenous AF9 ((**N**) $n = 2$ replicates) and its interaction with TFIID components.
**O** Immunoblotting analysis showing direct effect of PCAF-mediated acetylation of AF9 on its interaction with purified TFIID in vitro, following the experimental strategy shown in the left panel ($n = 1$ replicate). **P** Effect of PCAF knockdown on IR treatment-mediated acetylation of ectopically expressed AF9 and its concomitant interaction with TFIID components ($n = 2$ replicates).

Interestingly, we observed significantly reduced transcriptional activity post IR treatment at 1 hr when the AF9 oligomerization goes down, as well as its interaction with cognate interactors are also reduced (Fig. 3C, top panel for representative image and bottom panel for quantification). Further, qRT-PCR-based analysis of AF9-target gene expression also showed reduced level of all the target mRNAs at 2hrs time point that we have tested (Supplementary Fig. 3C). For the purpose of target mRNA analyses after IR treatment, we used 2hrs time point which could better reflect the overall effect of reduced nascent RNA transcription that we have observed at 1 hr time point. Therefore, for the purpose of detailed mechanistic understanding of functional regulation of AF9 in attaining transcriptional repression upon IR treatment, cells were treated with IR and assessed after 1 hr ((termed as IR(10 Gy)) for all of our subsequent assays, unless otherwise mentioned. Hence, we conclude that regulation of AF9 oligomerization and resultant interaction with cognate proteins could be a direct mechanism for temporal regulation of AF9 functions after exposure to DNA damage.

### Acetylation of AF9 strongly correlates with reduced interaction of AF9 with cognate interactors upon IR treatment
For deeper understanding of this regulation of AF9 oligomerization and concomitant factor association, we focused on acetylation since an earlier high-throughput study showed acetylation of AF9 protein upon DNA damage caused by UV exposure[28]. Interestingly, in our analyses, we also observed dynamic acetylation of AF9 after IR treatment, which peaks at 1 hr time point (Fig. 3D, compare lane 2 vs lane 3) and gradually decreases and goes back to normal by 8 hr time point (Fig. 3D, lanes 4, 5). Consistent with a role of DNA damage in enhancing AF9 acetylation, continued treatment of cells with camptothecin (a topoisomerase I inhibitor and causes double-strand break) also showed increased acetylation with increasing treatment time until it reaches to saturation (Supplementary Fig. 3D). This result, along with our observation using IR treatment as well as earlier result showing UV treatment-dependent enhanced AF9 acetylation indicates this (AF9 acetylation) as a universal response upon DNA damage within mammalian cells.

Consistent with our observation as shown in Fig. 3B, we also observed enhanced acetylation and concomitant reduced interaction of endogenous AF9 with TFIID as well as SEC components upon IR treatment (Fig. 3E, compare lane 4 vs lane 5 in the IP panel). Interestingly, the enhanced acetylation of AF9 is dependent on active transcription since prior treatment of cells with transcription inhibitor DRB (a P-TEFb inhibitor), abolishes this IR-induced acetylation, whereas parallel control experiment with DMSO treatment failed to show any effect (Supplementary Fig. 3E, compare lane 1 vs lanes 2, 3 and lane 4 vs lanes 5, 6). Thus, the overall effect of IR treatment on enhanced acetylation of AF9 is transcription-coupled. Overall, these data suggest a possible role of acetylation in regulation of AF9 functions through control of its oligomerization and concomitant TFIID association for overall transcriptional regulation upon IR treatment within mammalian cells.

### PCAF-mediated acetylation of AF9 both in vitro and in vivo within mammalian cells
Next, to identify the specific acetyl transferase that would be involved in AF9 acetylation within mammalian cells, our co-transfection experiments with multiple target acetyl transferases showed PCAF-specific acetylation of AF9 within mammalian cells (Fig. 3F, lane 3). Further analysis of co-transfection with PCAF only showed robust acetylation of ectopically expressed as well as endogenous AF9 within mammalian cells (Supplementary Fig. 3F,G, respectively). Consistent with these results, we have also observed strong PCAF-specific interaction with endogenous AF9 protein in presence of IR treatment (Supplementary Fig. 3H). Similar experiments failed to show much interaction with p300 and GCN5 in our experimental setup. To show a direct role of PCAF in acetylating AF9, we purified recombinant PCAF HAT domain (503–651 aa)[29], through its expression in bacterial system (Fig. 3G). Subsequent in vitro acetylation assay using purified GST-AF9 (Supplementary Fig. 2B) as well as PCAF HAT clearly showed dose-dependent acetylation of AF9 in presence of acetyl-CoA only (Fig. 3H, compare lanes 1–3 vs lanes 4–6). Therefore, all these results clearly show a direct role of PCAF in acetylation of AF9 both in vitro as well as in vivo within mammalian cells. To provide further evidence of role of PCAF in acetylating AF9 within mammalian cells, we generated several shRNA constructs for stable knockdown of PCAF (Supplementary Fig. 3I). Interestingly, knockdown of PCAF markedly reduced acetylation of endogenous AF9 (Fig. 3I, compare lane 5 vs lanes 6, 7). Consistent with a role of PCAF in regulation of acetylation, our immunoprecipitation assays of ectopically expressed (Supplementary Fig. 3J, lane 4) and endogenous AF9 (Supplementary Fig. 3K, lane 3) showed strong interaction of both these proteins with PCAF within mammalian cells. Overall, these results confirm that human AF9 is exclusively acetylated by PCAF both in vitro and in vivo within mammalian cells.

### PCAF-mediated acetylation reduces AF9 oligomerization leading to its reduced TFIID interaction
Since there is a strong correlation of AF9 acetylation and its oligomerization upon IR treatment (Fig. 3A, D, E), we wondered whether PCAF-mediated acetylation would regulate this oligomerization. In our initial analysis, PCAF-mediated acetylation showed reduced self-association between ectopically expressed AF9 proteins (Supplementary Fig. 3L, compare lane 5 vs lane 6). Consistent with a role of self-association in oligomerization, overexpression of PCAF markedly reduced endogenous AF9 oligomerization within mammalian cells

(Fig. 3J, compare lanes 3 vs lane 4), whereas, PCAF knockdown enhanced the overall oligomerization (Fig. 3K, compare lane 3 vs lane 4). Further, consistent with a role of AF9 oligomerization in YEATS domain-mediated interaction with TFIID, overexpression of PCAF markedly reduced specifically TFIID interaction and not SEC with ectopically expressed AF9 upon its acetylation (Fig. 3L, compare lane 2 vs lane 3). This observation is consistent with our earlier results, wherein, Poly-Ser domain-mediated oligomerization of AF9 affects only TFIID interaction and not SEC (Fig. 2). Opposite to the effect of overexpression of PCAF, its knockdown reduced acetylation level that in turn enhanced oligomerization leading to increased TFIID interaction with both ectopically expressed AF9 (Fig. 3M, compare lane 2 vs lane 3) as well as endogenous AF9 (Fig. 3N, compare lane 4 vs lane 5). Consistent with our observation within mammalian cells, our in vitro assay using the strategy (Fig. 3O, left panel) further provided direct evidence of role of PCAF-mediated acetylation in its reduced interaction with purified TFIID, when compared to control experiment without acetyl-CoA that failed to show any acetylation (Fig. 3O, right panel, compare lane 4 vs lane 5). Thus, based on all these experiments, we conclude that PCAF-mediated acetylation of AF9 reduces its oligomerization property, which, in turn, also reduces its interaction with TFIID alone and not the SEC components.

Consistent with a role for PCAF in enhancing AF9 acetylation upon IR treatment, knockdown of PCAF failed to show increased acetylation of ectopically expressed AF9 upon IR treatment within mammalian cells (Fig. 3P, compare lane 2 vs lane 3). This is also accompanied by enhanced TFIID interaction which is consistent with the role of acetylation in reducing AF9-TFIID interaction upon IR treatment. Thus, PCAF-mediated acetylation of AF9 plays a significant role in regulation of AF9 acetylation upon IR treatment within mammalian cells. This acetylation thus, in turn, reduces AF9 oligomerization and its interaction with TFIID for global downregulation of transcription upon IR treatment. Consistent with this hypothesis, PCAF KD cells failed to show efficient downregulation of global transcription, as measured through nascent RNA transcription by EU incorporation assays, when compared to control scramble (Scr) cells (Supplementary Fig. 3M, left panel for representative image and right panel for quantification).

### Lysine 339 (K339) of AF9 is critically targeted for PCAF-mediated acetylation within mammalian cells

For addressing the specific residue(s) within AF9 that would be targeted for this acetylation, we targeted lysine residues present within the 329–448 region since our initial domain analyses showed that a deletion between C-terminal 329 to 448 aa residues almost abolished PCAF-mediated acetylation within mammalian cells (Supplementary Fig. 4A, compare lane 4 vs lane 5). We also targeted the fragment harboring 209–328 amino acids which also showed some acetylation signal in our domain analyses (Supplementary Fig. 4A, lane 5). Introduction of Lysine to Arginine (K to R) mutation, for keeping the similar charge as well as amino acid structure within the protein fragment, showed that the fragment containing K339R mutant in combination with other mutations (Fig. 4A, lane 9) showed a dramatic effect on overall PCAF-mediated acetylation, whereas, other mutations showed either very modest or no effect (Fig. 4A, compare lane 9 with other lanes). Subsequent analysis with K339R mutant alone showed a marked effect on overall PCAF-mediated acetylation within mammalian cells (Fig. 4B, compare lane 3 vs lane 4).To investigate the specific role of PCAF-mediated acetylation of K339 residue of AF9 and its physiological relevance, we generated a polyclonal antibody specific to the acetylated K339 residue (Supplementary Fig. 4B). The generated antibody is very specific to the acetylated K339 (K339Ac) residue since in our immunoblotting assay, the antibody did not detect much acetylation in the presence of PCAF when we used the AF9(K339R) mutant (Fig. 4C, compare lane 3 vs lane 4). Consistently, our immunoblotting analyses by using the generated K339Ac-specific antibody have further

validated that ectopic expression of PCAF enhanced K339 acetylation of endogenous AF9 (Supplementary Fig. 4C, compare lane 1 vs lane 2) and its knockdown reduced overall acetylation level within mammalian cells (Fig. 4D). Therefore, we conclude that K339 of AF9 is the major site for PCAF-mediated acetylation within mammalian cells.

To further understand specific role of PCAF-mediated acetylation of K339 residue in regulation of AF9 functions, we generated a K339Ac mimic mutant (K339Q). In our initial analyses, the K339Q mutant showed reduced oligomerization potential (Fig. 4E, compare lane 3 vs lane 4), like the one observed for AF9 acetylation in presence of over-expressed PCAF within mammalian cells (Fig. 3J). Further, we also observed reduced interaction of K339Q mutant of AF9 with TFIID and not the SEC within cells (Fig. 4F, compare lane 2 vs lane 3). This observation is similar to the effects of PCAF-mediated acetylation on oligomerization and SEC interaction (Fig. 3L, M). Also, consistent with reduced interaction of TFIID with AF9(K339Q) mutant, while re-expression of AF9(WT) in AF9 KD cells restored target gene expression, the K339Q mutant failed to do so (Fig. 4G). Further, ChIP analyses of factor binding on some of these target genes showed restoration of recruitment of target factors upon expression of AF9(WT) in the AF9 KD cells (Fig. 4H, compare EV vs WT) and release of paused Pol II from promoter-proximal region. However, owing to reduced oligomerization and TFIID interaction, the K339Q mutant failed to restore factor recruitment and release of paused Pol II as observed in AF9 KD cells (Fig. 4H, compare EV vs K339Q). Pausing index analysis also showed that while re-expression of AF9(WT) reduced the pausing, similar expression of AF9(K339Q) failed to do so ((Fig. 4H, right panel, compare EV vs AF9(WT) and EV vs AF9(K339Q)). Similarly, we have observed restoration of proliferation and colony formation ability of AF9 KD cells re-expressing AF9(WT) as compared to control EV ((Supplementary Fig. 4D, E, compare EV vs AF9 (WT)). In contrast, the knockdown cells expressing AF9 (K339Q) mutant failed to do so((Fig. Supplementary Fig. 4D, E, compare AF9(WT) vs AF9 (K339Q)). These results clearly establish a role that PCAF-mediated acetylation of K339 residue critically regulates AF9 oligomerization and subsequent interaction with TFIID for regulation of factor recruitment on target genes for their expression and subsequent regulation of proliferation and colony formation of mammalian cells.

### Exposure to genotoxic stress causes PCAF-mediated AF9 acetylation at K339 residue to inhibit expression of global target genes

Since our earlier analyses showed strong correlation between IR-induced PCAF-mediated AF9 acetylation and concomitant reduced interaction with TFIID leading to global transcriptional downregulation (Fig. 3), we wondered whether the K339 residue would be a critical target for this regulation. Interestingly, the AF9(K339R) mutant failed to show enhanced K339 acetylation (Fig. 4I, compare lane 3 vs lane 4) and reduced oligomerization upon IR treatment (Fig. 4J, compare lane 3 vs lane 4), which subsequently did not show any effect on TFIID interaction upon IR treatment (Fig. 4K, compare lane 7 vs lane 8). In fact, the K339R mutant showed enhanced interaction than that of AF9(WT) upon IR treatment. Consistent with our observation with ectopically expressed AF9, endogenous AF9 also showed enhanced acetylation at K339 residue at 1 hr post IR treatment (Fig. 4L, lane 7) when global transcriptional downregulation within mammalian cells is maximum as observed through EU incorporation assay (Fig. 3C). This is also accompanied by reduced interaction with TFIID components (Fig. 4L, compare lane 6 vs lane 7) at 1 hr post IR treatment. Further, our immunofluorescence analysis using K339Ac-specific antibody also showed significant increase in overall acetylation at K339 residue (Supplementary Fig. 4F, left panel for representative image and right panel for quantification). Further, emphasizing a role for PCAF in IR treatment-dependent K339 acetylation, its knockdown dramatically abolished overall K339 acetylation upon IR treatment (Supplementary

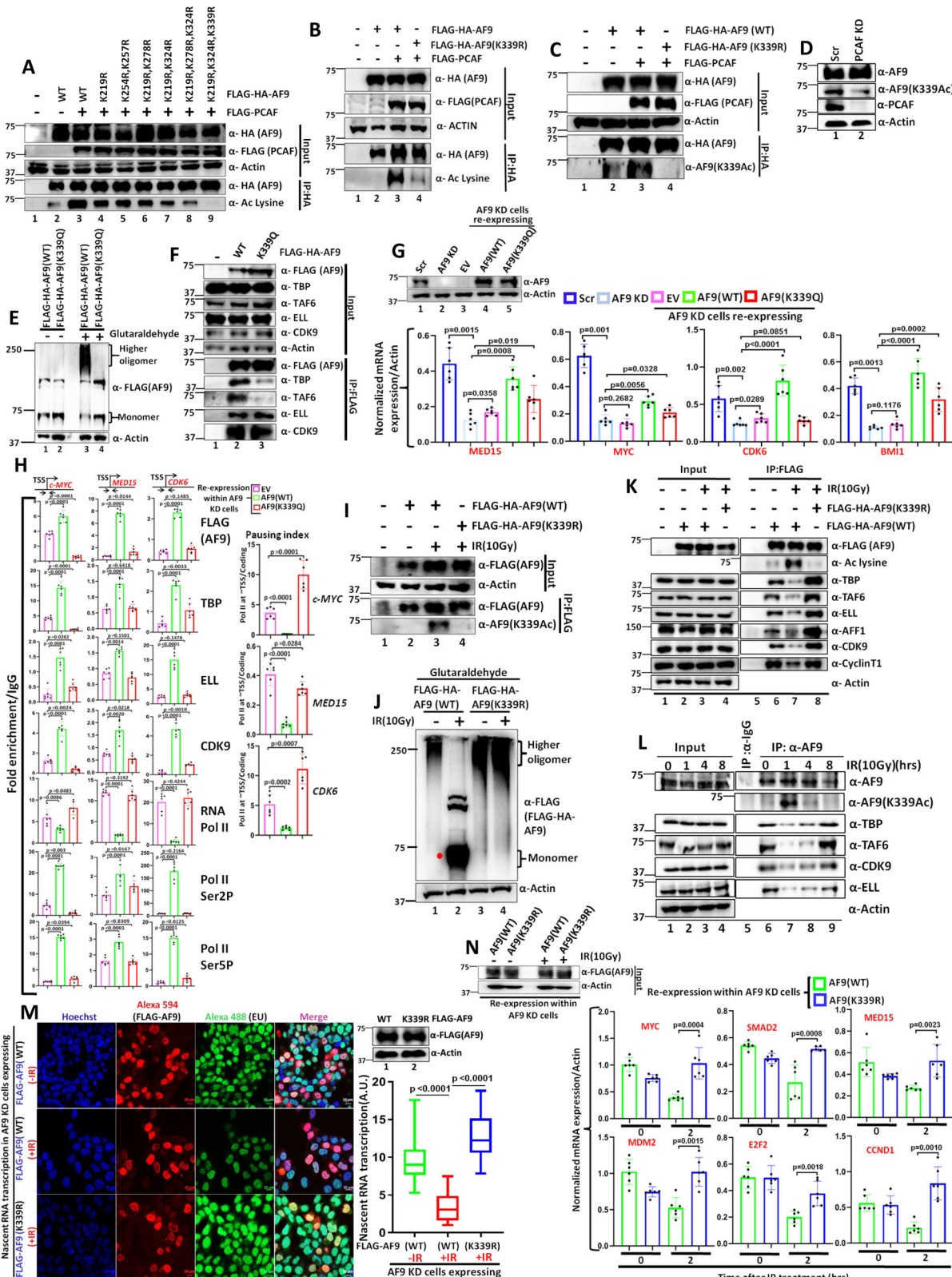

Fig. 4G, left panel for representative image and right panel for quantification).

Next, we addressed whether IR treatment-dependent K339 acetylation would be a key mechanism for global transcriptional shutdown since this acetylation has been shown to impair target gene activation under normal cellular growth (Fig. 4G,H). For this purpose, we re-expressed the AF9(WT) and K339R mutant in AF9 KD cells (Fig. 4M,

right panel) and used for subsequent EU incorporation assay for addressing nascent RNA transcription. Interestingly, expression of AF9(WT) in the AF9 KD cells showed significant transcriptional activity as assayed by EU incorporation assay under normal cellular growth (Fig. 4M, −IR top left panels for representative image and right panel for quantification). However, upon IR treatment, we observed significant downregulation of nascent RNA transcription activity in the

**Fig. 4 | PCAF-mediated acetylation of AF9 at lysine 339 (K339) residue reduces its oligomerization potential and TFIID interaction for downregulation of global transcription during genotoxic stress within 293 T cells. A** Acetylation of ectopically expressed AF9 and its mutant derivatives by PCAF within mammalian cells (*n* = 2 replicates). **B** Acetylation of AF9(WT) and AF9(K339R) by PCAF within mammalian cells (*n* = 3 replicates). **C** Immunoblotting analysis showing PCAF-mediated acetylation of AF9 at K339 residue using AF9(K339Ac)-specific antibody (*n* = 1 replicate). **D** Effect of PCAF KD on acetylation of endogenous AF9 at K339 residue (*n* = 2 replicates). **E, F** Oligomerization of AF9(WT) and AF9 (K339Q) ((**E**) *n* = 1 replicate) and its effect on interaction with TFIID and SEC components ((**F**) *n* = 1 replicate). **G, H** qRT-PCR ((**G**) *n* = 2 replicates) and ChIP analyses ((**H**) *n* = 2 replicates) showing effect of re-expression of AF9(WT) and AF9(K339Q) proteins in stable AF9 KD cells on target mRNA expression (**G**) and recruitment of indicated target factors at the promoter-proximal region of selected AF9-target genes (**H**). **I** Immunoblotting analysis showing IR-induced acetylation of AF9(WT) and AF9(K339R) at K339 residue, using AF9(K339Ac) antibody (*n* = 1 replicate).

**J** Reduced oligomerization of AF9(WT) as compared to AF9 (K339R) upon IR(10 Gy) treatment (*n* = 2 replicates). **K** Interaction of AF9 (K339R) mutant with TFIID and SEC components upon IR(10 Gy) treatment (*n* = 3 replicates). **L** Effect of IR treatment on dynamic acetylation of endogenous AF9 at K339 residue and its concomitant effect on interaction with TFIID and SEC components (*n* = 2 replicates). **M** Nascent RNA transcription analysis showing enhanced global transcription ability in stable AF9 KD cells expressing AF9(K339R) when compared to AF9(WT) after 1 hr of IR treatment. The boxes represent median and quartiles and value ranges of 25–75 and 10–90%. Upper and lower hinges extend to the largest and smallest data points (*n* = 100 cells). This experiment was done once. **N** Effect of re-expression of indicated proteins in stable AF9 KD cells on mRNA expression of indicated AF9-target genes at 2hrs after IR treatment (*n* = 2 replicates). For this figure, the data showing RNA and ChIP analyses by qRT-PCR, the error bar represents mean ± SD, and statistical analyses were performed using one-tailed Student's *t* test. *p* values for each experimental data is mentioned on the bar diagram.

AF9 KD cells wherein the AF9(WT) protein has been re-expressed (Fig. 4M, +IR middle left panels for representative image and right panel for quantification). Importantly, re-expression of AF9(K339R) mutant in KD cells completely failed to show any downregulation of expression (Fig. 4M, +IR lower left panels for representative image and right panel for quantification). Further, analyses of AF9-target gene expression showed downregulation of expression at 2hrs post IR treatment in cells expressing AF9(WT) ((Fig. 4N, compare mRNA expression at 0 hr vs 2hrs in cells expressing AF9(WT)). However, the AF9(K339R)-expressing knockdown cells failed to show downregulation of target gene expression at 2hrs after IR treatment. Consistent with reduced interaction of AF9 with both TFIID as well as SEC components upon IR treatment (Fig. 2), we also observed reduced recruitment of all these factors at the promoter-proximal regions of all these target genes in cells expressing AF9(WT) resulting in reduced level of Pol II Ser2 and Ser5 phosphorylation and enhanced pausing (Supplementary Fig. 4H). However, cells expressing the AF9(K339R) mutant of AF9 that fully retained both TFIID as well as SEC interaction (Fig. 4K), also showed efficient recruitment of all these factors leading to optimal level of Pol II Ser2 and Ser5 phosphorylation and reduced pausing that correlates well with expression of target genes (Supplementary Fig. 4H).

Thus, all these data clearly establish a role for PCAF-mediated acetylation of K339 residue of AF9 in temporal regulation of gene expression upon IR treatment, through reduced oligomerization and impaired TFIID interaction. Further, consistent with the role of AF9 in acetylation-dependent global transcriptional downregulation for optimal repair response and cell survival upon IR treatment, the AF9 KD cells expressing K339R mutant showed reduced proliferation potential than the WT after IR treatment (Supplementary Fig. 4I). Interestingly, unlike our observation of acetylation mimic mutant (K339Q), as shown in Fig. 4H, that shows reduced AF9 recruitment onto target genes under normal cellular growth, IR-induced acetylation of AF9(WT) fails to show this reduction onto target genes despite its loss of interaction with other components. These observations suggest the presence of additional mechanism(s) in retaining AF9 onto chromatin upon IR treatment that may have functional implications beyond the regulation of transcription alone (see below).

## DNA-PKc-mediated phosphorylation at Serine 395 (S395) residue of AF9 regulates its SEC interaction during genotoxic stress

Our earlier results showed that along with TFIID, AF9 also shows reduced interaction with SEC components upon IR treatment within mammalian cells (Fig. 3B, Supplementary Figs. 3B and 4L). However, PCAF-mediated acetylation specifically reduces TFIID interaction with AF9, without impairing its interaction with SEC components. What is/are the underlying mechanism(s) of reduced interaction of AF9 with SEC components upon exposure to genotoxic stress? In response to

DNA damage, multiple phosphoinositide-3-like kinases (PIKKs), including ATR, ATM, and DNA-PKc, phosphorylate target substrates having common motif (S/T-Q)[30]. We initially checked whether AF9 protein would be subjected to phosphorylation by PIKKs upon IR treatment at different time points. Surprisingly, we observed enhanced phosphorylation of both ectopically expressed (Fig. 5A, lane 2) and endogenous AF9 (Fig. 5B, lane 7) at S/T-Q motif at 1 hr post IR treatment that coincides with acetylation signal as well. We used a specific antibody that recognizes phosphorylation at S/T-Q motif for our immunoblotting analyses. Interestingly, at this same time point, we also observed concomitant reduction of interaction with ectopic AF9 with SEC components, e.g., CDK9 and ELL (Fig. 5C, compare lane 2 vs lane 3). Immunoprecipitation of endogenous AF9 also showed similar results (Fig. 5D, compare lane 4 vs lane 5).

To identify specific PIKK involved in AF9 phosphorylation upon DNA damage, we employed mass spectrometry analysis of AF9-immunoprecipitated proteins from nuclear extract of a stable cell line that ectopically express FLAG-HA-AF9[18]. Interestingly, we observed strong interaction of AF9 with DNA-PKc within 293 T cells by mass spectrometry analysis (Supplementary Fig. 5A), which was further confirmed by our immunoprecipitation analysis (Fig. 5E, lane 2). Further analyses also showed strong DNA-PKc-specific interaction with both ectopically expressed as well as endogenous AF9 proteins (Supplementary Fig. 5B, C) thus substantiating a role for DNA-PKc in functional regulation of AF9 protein within mammalian cells. Consistent with a role of DNA-PKc in this phosphorylation, treatment of cells with DNA-PKc-specific inhibitor (NU7441) showed reduced phosphorylation of ectopically expressed (Fig. 5F, compare lane 3 vs lane 4) as well as endogenous AF9(Fig. 5G, compare lane 6 vs lane 7) and unimpaired interaction of endogenous AF9 with SEC components (Fig. 5G, compare lane 6 vs lane 7). Collectively, these data suggested a role for DNA-PKc in IR treatment-dependent phosphorylation of AF9 that is directly associated with reduced SEC interaction.

For identifying the critical residue within AF9 that would be targeted for DNA-PKc-mediated phosphorylation, we focused on single S/T-Q motif present at Serine 395 position within AF9. Introducing a mutation in this residue (S395A) clearly showed abolition of phosphorylation of AF9 upon IR treatment (Fig. 5H, compare lane 3 vs lane 4). More importantly, loss of phosphorylation also caused retention of SEC interaction which otherwise is lost for AF9(WT) upon IR treatment (Fig. 5H, compare lane 2 vs lanes 3, 4). An earlier report has shown a role for UV treatment-dependent ATM-mediated ENL phosphorylation at two conserved ATM phosphorylation sites in enhancing its interaction with PRC1 complex component BMI1 for transcriptional repression[12]. Consistent with that observation, we have also observed enhanced BMI interaction with AF9 upon IR treatment that correlates well with its phosphorylation (Fig. 5I, compare lane 2 vs lane 3). However, a loss of this phosphorylation also caused reduced BMI1 interaction (lane 4).

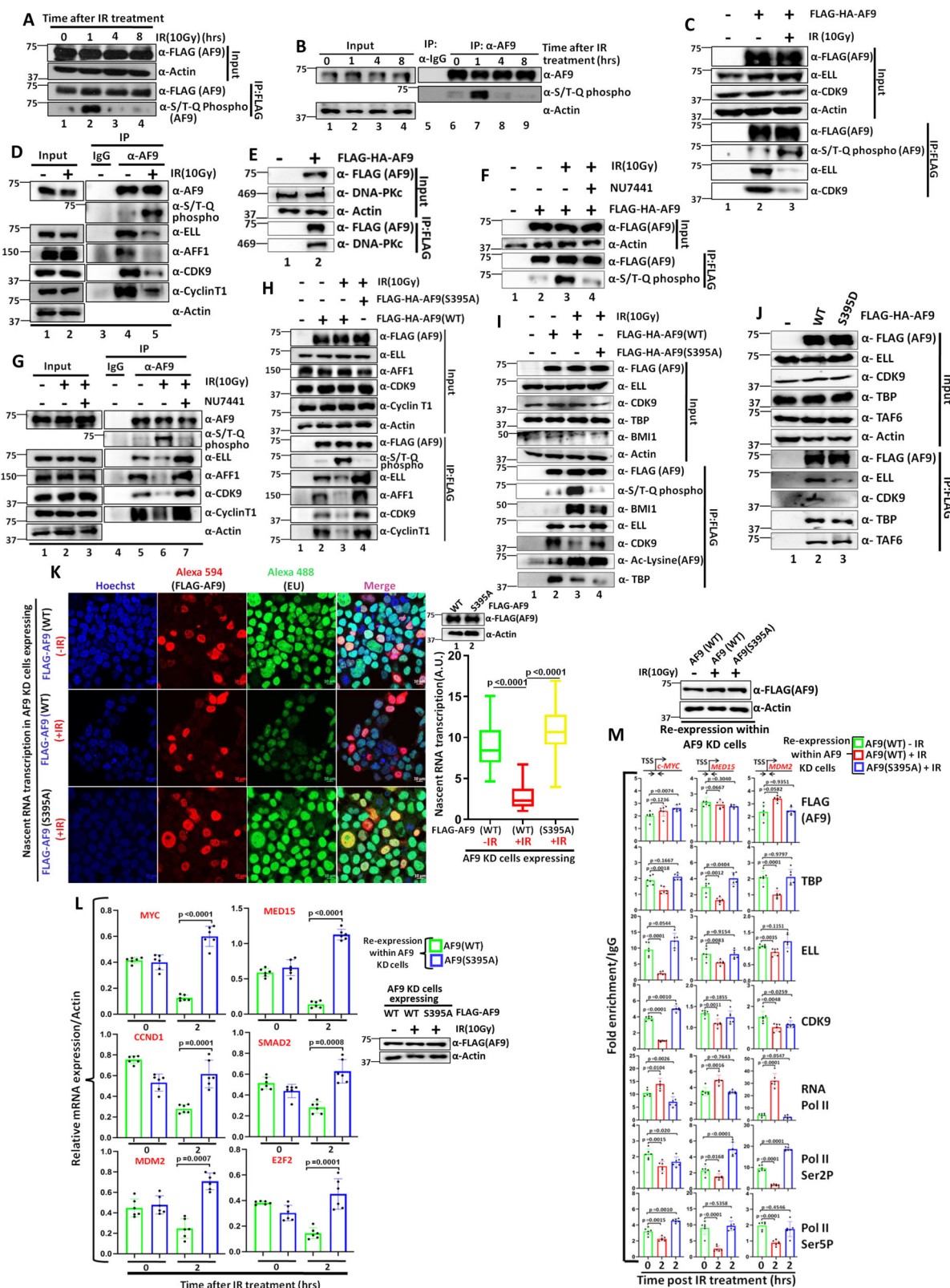

Interestingly, the overall effect of DNA-PKc-mediated phosphorylation at S395 residue is specific to SEC interaction only, since in the same assay, we failed to observe any effect of S395A mutation on enhanced AF9 acetylation and concomitant reduction in TFIID interaction (Fig. 5I, compare lane 2 vs lanes 3, 4). Consistent with a direct role of AF9 phosphorylation at S395 residue in reducing SEC interaction, a phosphorylation mimic mutant (S395D) also efficiently reduced AF9

interaction with SEC component alone without affecting TFIID interaction under normal cellular growth (Fig. 5J, compare lane 2 vs lane 3). Thus, all these data clearly showed a role for IR treatment-dependent DNA-PKc-mediated phosphorylation of AF9 at S395 residue in reducing SEC interaction that may have an implication in efficient global transcriptional downregulation. Indeed, consistent with these hypotheses, AF9 KD cells re-expressing mutant AF9(S395A) fails to efficiently

**Fig. 5 | DNA-PKc-mediated phosphorylation of AF9 at Serine 395 (S395) residue reduces its interaction with SEC components for transcriptional downregulation during genotoxic stress within 293 T cells. A, B** IR-induced dynamic phosphorylation of ectopically expressed AF9 ((**A**) $n = 3$ replicates) and endogenous AF9 ((**B**) $n = 1$ replicate). **C, D** Enhanced phosphorylation of ectopically expressed AF9 ((**C**), $n = 3$ replicates) and endogenous AF9 ((**D**) $n = 2$ replicates), and its concomitant reduced interaction with SEC components upon IR(10 Gy) treatment. **E** Interaction of ectopically expressed AF9 protein with endogenous DNA-PKc within mammalian cells ($n = 3$ replicates). **F, G** Effect of treatment of cells with NU7441(2 μM) on IR(10 Gy) treatment-dependent phosphorylation of ectopically expressed AF9 ((**F**) $n = 2$ replicates) and endogenous AF9 ((**G**) $n = 3$ replicates) and its concomitant interaction with SEC components **G. H** Immunoblotting analysis showing defective phosphorylation and unaltered SEC interaction in cells expressing phosphorylation-defective AF9 mutant (S395A) when compared to AF9(WT), upon IR(10 Gy) treatment ($n = 3$ replicates). **I** Effect of AF9(S395A) mutant on IR(10 Gy)-dependent phosphorylation and concomitant acetylation and its effect on interaction with SEC and TFIID components ($n = 1$ replicate). **J** Effect of AF9

phosphorylation mimic mutant (S395D) on its interaction with TFIID and SEC components ($n = 1$ replicate). **K** Nascent RNA transcription analysis showing enhanced global transcriptional activity in cells expressing AF9(S395A) mutant, when compared to AF9(WT) upon IR(10 Gy) treatment. The boxes represent median and quartiles and value ranges of 25–75 and 10–90%. Upper and lower hinges extend to the largest and smallest datapoints ($n = 100$ cells). This experiment was done once. **L** qRT-PCR analysis showing effect of re-expression of AF9(WT) and AF9(S395A) proteins in stable AF9 KD cells on mRNA expression of indicated AF9-target genes at 2hrs after IR(10 Gy) treatment ($n = 2$ replicates). **M** ChIP analysis showing recruitment of indicated factors at the promoter-proximal regions of AF9-target genes upon re-expression of AF9(S395A) in stable AF9 KD cells when compared to AF9(WT) at 2hrs after IR(10 Gy) treatment ($n = 2$ replicates). For this figure, the data showing RNA and ChIP analyses by qRT-PCR, the error bar represents mean ± SD and statistical analyses were performed using one-tailed Student's $t$ test. $p$ values for each experimental data is mentioned on the bar diagram.

---

downregulate nascent RNA transcription than the one in which AF9(WT) has been re-expressed (Fig. 5K, compare nascent RNA transcription after IR treatment in presence of indicated AF9-expressing plasmids). Further, analyses of AF9-target gene expression by qRT-PCR also showed efficient downregulation of expression of target genes in the AF9 KD cells re-expressing the AF9(WT) (Fig. 5L). However, knockdown cells re-expressing AF9(S395A) mutant failed to show this downregulation of expression (Fig. 5L). In fact, upon IR treatment, we observed enhanced expression than the basal level of all the target genes that we have tested. Consistent with reduced RNA expression, subsequent ChIP analyses further showed reduced recruitment of all the target factors in cells expressing AF9(WT) upon IR(10 Gy) treatment (Fig. 5M). However, cells expressing AF9(S395A) mutant failed to show this reduction, an observation that is consistent with its interaction with cognate partners upon IR treatment. Once again, similar to our observation with AF9(K339R) mutant, in this case as well, we observed similar recruitment of AF9 both in the WT as well as S395A mutant suggesting an additional role of AF9 beyond transcription in the overall functional regulation.

## Acetylation of AF9 at K339 residue enhances its interaction for DNA-PKc for downstream phosphorylation at S395 residue

We were interested in addressing the cross-talk between PCAF-mediated acetylation and DNA-PKc-mediated phosphorylation of AF9 in overall regulation since both these modifications peak at the same time point after IR treatment. Intriguingly, we observed enhanced interaction of DNA-PKc with ectopically expressed (Fig. 6A, compare lane 2 vs lane 3) as well as endogenous AF9 (Fig. 6B, compare lane 4 vs lane 5) upon IR treatment. Immunoprecipitation of endogenous AF9 also showed strong correlation of enhanced interaction of DNA-PKc with increased acetylation at K339 residue upon IR treatment (Fig. 6B). This observation raised the possibility of IR-dependent PCAF-mediated acetylation in regulation of AF9 interaction with DNA-PKc. Indeed, knockdown of PCAF markedly reduced IR-dependent enhanced acetylation as well as concomitant DNA-PKc interaction with ectopically expressed (Supplementary Fig. 6A, compare lane 3 vs lane 4) as well as endogenous AF9 (Fig. 6C, compare lane 6 vs lane 7) that is well correlated with reduced phosphorylation of ectopic (Supplementary Fig. 6B, compare lane 3 vs lane 4) and endogenous AF9 (Fig. 6D, compare lane 6 vs lane 7). Interestingly, consistent with a role of DNA-PKc-mediated phosphorylation of AF9 in reducing SEC interaction, we have failed to observe any reduced SEC interaction with endogenous AF9 in PCAF KD cells upon IR treatment (Fig. 6E, compare lane 6 vs lane 7). Further, to show a role for PCAF-mediated AF9 acetylation at K339 residue in regulation of interaction with DNA-PKc, the acetylation-defective AF9(K339R) mutant failed to show enhanced interaction with DNA-PKc upon IR treatment, which otherwise is observed for AF9(WT) (Fig. 6F, compare lane 2 vs lane 3 and lane 3 vs

lane 4). Further, consistent with a role for AF9 acetylation at K339 residue in regulating DNA-PKc-mediated phosphorylation in regulation of SEC recruitment upon genotoxic stress, the AF9(K339R) mutant fails to show enhanced acetylation and concomitant phosphorylation at S395 residue, when compared to AF9(WT) (Fig. 6G, compare lane 3 vs lane 4). This also results in failure to show reduction of TFIID as well as SEC interaction with AF9 upon IR(10 Gy) treatment (Fig. 6G, compare lane 3 vs lane 4).

Based on these interaction data, it can be hypothesized that the enhanced interaction of DNA-PKc upon AF9 acetylation at K339 could be result of two possible effects in which a) acetylation at K339 residue directly enhances DNA-PKc interaction or b) acetylation-dependent monomerization of AF9 enhances overall DNA-PKc interaction. To distinguish between these possibilities, we checked the DNA-PKc interaction using Poly-SerΔ mutant of AF9 that predominantly forms monomer. Contrary to enhanced interaction, this construct showed reduced association with DNA-PKc compared to WT upon IR treatment (Supplementary Fig. 6C, compare lane 4 vs lane 5). Interestingly, the Poly-SerΔ mutant also showed impaired acetylation with PCAF, as shown in Supplementary Fig. 6D (compare lane 5 vs lane 6). Thus, the enhanced interaction of DNA-PKc is solely dependent on acetylation and not the acetylation-dependent monomerization since monomerized AF9 Poly-SerΔ construct failed to show any enhancement of interaction. Overall, these results clearly showed a role for genotoxic stress-dependent PCAF-mediated acetylation of AF9 at K339 residue in regulation of its interaction with DNA-PKc for downstream phosphorylation at S395 residue for further regulation of SEC interaction for overall temporal transcriptional downregulation.

## IR-mediated AF9 acetylation-dependent enhanced DNA-PKc recruitment also results in enhanced interaction with components of Ku complex

Since DNA-PKc also plays a key role in regulation of DNA repair through non-homologous end joining (NHEJ) pathway by recruiting components of Ku complex, we envisaged that the enhanced DNA-PKc interaction upon AF9 acetylation could as well be a mechanism for coupling global transcriptional inhibition and repair of damaged DNA through enhanced interaction and possibly, increased recruitment of components of DNA-PKc-interacting repair machineries such as Ku complex[30]. Consistent with this hypothesis, our initial analysis showed enhanced interaction of ectopically expressed (Supplementary Fig. 6E, compare lane 2 vs lane 3) and endogenous AF9 (Fig. 6H, compare lane 4 vs lane 5) with Ku70 and Ku80 proteins (components of Ku complex) along with DNA-PKc upon IR treatment. Further, to substantiate a role of AF9 acetylation at K339 residue in regulation of these interactions, immunoprecipitation analysis of ectopically expressed AF9 (K339R) protein in AF9 KD cells showed markedly impaired DNA-PKc

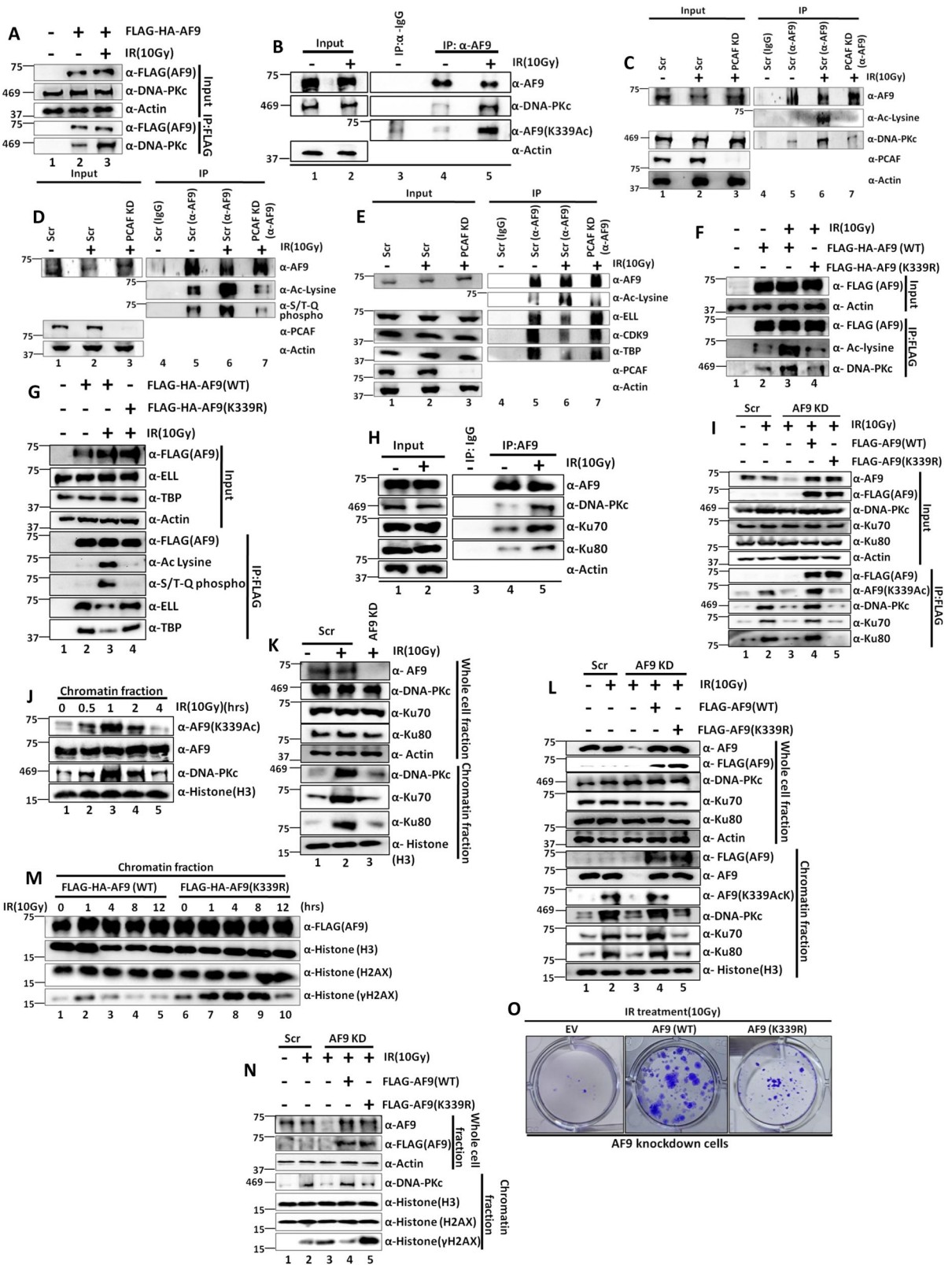

interaction along with downstream interaction with Ku70 and Ku80 proteins when compared to control AF9(WT) (Fig. 6I, compare lane 4 vs lane 5). These results clearly indicate that, besides its role in regulation of SEC interaction through phosphorylation, PCAF-mediated AF9 acetylation-dependent enhanced DNA-PKc interaction with AF9 may have additional role in recruitment of Ku complex in regulation of repair of damaged DNA.

## IR-dependent PCAF-mediated acetylation of AF9 at K339 residue also increases DNA-PKc and Ku complex recruitment on chromatin for efficient repair of damaged DNA

Since we have observed enhanced interaction of DNA-PKc and concomitant Ku70 and Ku80 proteins by PCAF-mediated acetylation of AF9 upon IR treatment, we wondered, whether, this could be a mechanism for efficient repair of damaged DNA through their

**Fig. 6 | AF9 K339 acetylation-dependent DNA-PKc recruitment onto chromatin facilitates phosphorylation at S395 residue as well as recruits Ku complex for DNA damage repair within 293 T cells. A, B** Immunoblotting analysis showing increased interaction of ectopically expressed AF9 ((**A**), $n = 2$ replicates) and endogenous AF9 ((**B**) $n = 2$ replicates) with DNA-PKc and its concomitant acetylation at K339 residue within mammalian cells upon exposure to IR(10 Gy) **B C, D** Effect of PCAF KD on IR(10 Gy) treatment-dependent acetylation of endogenous AF9 and concomitant DNA-PKc interaction ((**C**) $n = 1$ replicate) and phosphorylation at S395 residue ((**D**) $n = 1$ replicate). **E** PCAF KD fails to show reduced interaction of TFIID and SEC components with endogenous AF9 within mammalian cells upon IR(10 Gy) treatment ($n = 1$ replicate). **F** Effect of AF9(K339R) on IR treatment-dependent enhanced interaction with DNA-PKc within mammalian cells ($n = 2$ replicates). **G** Effect of AF9(K339R) mutant on IR treatment-dependent acetylation and concomitant phosphorylation-dependent association with TFIID and SEC components ($n = 2$ replicates). **H** Enhanced interaction of endogenous AF9 with DNA-PKc and concomitant Ku complex components within mammalian cells upon IR(10 Gy) treatment ($n = 2$ replicates). **I** Effect of acetylation of AF9 at K339 residue on IR(10 Gy) treatment-dependent enhanced interaction with DNA-PKc and Ku complex components within mammalian cells ($n = 2$ replicates). **J** Chromatin association of acetylated AF9(K339Ac) and concomitant DNA-PKc at different time points after IR treatment ($n = 2$ replicates). **K** Effect of stable AF9 KD on association of DNA-PKc and Ku complex components onto chromatin after IR(10 Gy) treatment ($n = 1$ replicate). **L** Effect of re-expression of AF9(WT) and AF9(K339R) in stable AF9 knockdown cells on association of DNA-PKc and Ku complex components onto chromatin upon IR(10 Gy) treatment ($n = 2$ replicates). **M** Effect of overexpression of AF9(WT) and AF9(K339R) mutant on repair of damaged DNA at indicated time points after IR(10 Gy) treatment ($n = 2$ replicates). **N, O** Effect of re-expression of AF9(WT) and AF9(K339R) in stable AF9 knockdown cells on overall DNA damage (**N**) $n = 1$ replicate) and colony forming potential after IR(10 Gy) treatment (**O**) ($n = 3$ replicates).

recruitment on chromatin. Our initial analysis showed that although the level of chromatin-associated AF9 protein does not change over time after IR treatment (as indicated in Fig. 6J as well as ChIP analyses in Supplementary Figs. 4H and 5 M), the overall level of acetylated AF9 at K339 residue (K339Ac) is changed, wherein, in the initial hours after IR treatment, enhanced K339Ac signal is observed and is peaked at 1 hr time point (Fig. 6J). However, at 4hrs time point, the level of K339Ac is reduced. Interestingly, we have also observed similar recruitment/ presence of DNA-PKc onto the chromatin which coincides with the level of acetylated AF9 at K339 residue (Fig. 6J). Consistent with a role of AF9 in regulation of recruitment of DNA-PKc and associated Ku complex for efficient repair of damaged DNA, our initial analysis showed reduced recruitment of these factors onto chromatin upon IR treatment in AF9 KD cells when compared to scramble control (Fig. 6K, compare lane 2 vs lane 3). Further, overexpression of acetylation-defective AF9 mutant (K339R) also showed reduced DNA-PKc recruitment onto chromatin upon IR treatment (Supplementary Fig. 6F). Interestingly, while ectopic expression of AF9(WT) in AF9 KD cells restores impaired DNA-PKc and Ku complex recruitment onto chromatin (Fig. 6L, compare lane 3 vs lane 4) upon IR treatment, the acetylation-defective AF9(K339R) mutant failed to do so (compare lane 4 vs lane 5). Thus, all these experiments clearly demonstrate a role for PCAF-mediated acetylation of AF9 at K339 residue in regulation of recruitment of DNA-PKc and downstream Ku complex onto chromatin potentially for repair of damaged DNA.

Next, since DNA-PKc and Ku complex are involved in repair of damaged DNA after genotoxic stress that primarily involves NHEJ pathway, we wondered whether AF9 acetylation-mediated recruitment of these complexes onto chromatin can have additional role in repair of damaged DNA. Our initial analyses showed enhanced presence of γ-H2AX signal in the AF9 KD cells even under normal cellular growth (Supplementary Fig. 6G, compare lane 1 vs lane 2). Further, over-expression of acetylation-defective AF9(K339R) mutant showed prolonged presence of γ-H2AX signal within mammalian cells upon IR treatment than the AF9(WT) (Fig. 6M, compare lanes 1–5 vs lanes 6–10). Consistent with the restoration of recruitment of DNA-PKc and Ku complex onto chromatin (Fig. 6L), while re-expression of AF9(WT) in knockdown cells reduced overall γ-H2AX signal, the acetylation-defective K339R mutant failed to do so (Fig. 6N, compare lane 3 vs lanes 4, 5). Further, kinetic analyses showed efficient repair by 4hrs of time in AF9 KD cells expressing AF9(WT) (Supplementary Fig. 6H, compare lane 2 vs lane 3). However, the acetylation-defective K339 mutant failed to do so (compare lane 8 vs lane 9). The overall effect is transcription-coupled, since reducing transcription by treating the cells with DRB markedly impaired overall repair potential in presence of ectopically expressed AF9 (Supplementary Fig. 6H, compare lanes 1–3 vs lanes 4–6). Consistent with this mechanism of repair of damaged DNA, re-expression of the acetylation-defective AF9(K339R) mutant in AF9 KD cells shows markedly impaired colony formation

potential after IR treatment when compared to control AF9(WT) (Fig. 6O). Thus, based on all these evidences, we conclude that PCAF-mediated acetylation of AF9 at K339 is important for enhanced interaction and recruitment of DNA-PKc and Ku complex components onto chromatin for efficient repair of damaged DNA. Thus, AF9/DNA-PKc axis has two important functions to regulate a) AF9-mediated interaction of SEC components for global transcriptional repression and b) recruit DNA-PKc and Ku complex onto chromatin for aiding repair of damaged DNA involving NHEJ pathway.

## HDAC5 is the key deacetylase that regulates dynamic acetylation level of AF9 upon IR treatment

Since we have observed PCAF-mediated acetylation of AF9 in temporal regulation of IR treatment-dependent global transcriptional down-regulation and subsequent DNA repair involving DNA-PKc and Ku complex, we were also interested in addressing the downstream events more specifically restart of transcription after repair of damaged DNA. As shown in Fig. 3C, as early as 1 hr time point after cells are treated with IR, we observed global downregulation of transcription. However, transcription is slowly resumed and by 8hrs time point onwards, full transcriptional potential is achieved as observed through nascent RNA transcription analyses. Since PCAF-mediated acetylation plays an important role in AF9-mediated downregulation of global transcription, we addressed the overall dynamics of AF9 acetylation and its interaction with cognate interactors at different time points after IR treatment. As shown in Fig. 7A, the initial enhanced acetylation observed at 1 hr time point, comes back to basal level by 8hrs time point. The AF9 acetylation-dependent reduced interaction with other key factors is also markedly restored by 8hrs time point onwards. Similar results are also observed when endogenous AF9 is immuno-precipitated (Fig. 4L). These results thus suggest that acetylation of AF9 is temporally regulated in which function(s) of unknown deace-tylase(s) could play a key role in overall regulation.

To address the role of unknown deacetylase(s) in this overall functional regulation, we focused on regulation of AF9 acetylation by histone deacetylases (HDACs) that play important roles in DNA damage-dependent response within mammalian cells[31]. In our initial analyses, we observed markedly stronger interaction of ectopically expressed AF9 with HDAC4 and HDAC5 when they are co-expressed within mammalian cells (Fig. 7B, compare lanes 6, 7 with other lanes). Subsequent co-expression analyses showed reversal of PCAF-mediated acetylation of AF9 in presence of HDAC4 and HDAC5 (Fig. 7C, compare lane 3 vs lane 4 and lane 5 vs lane 6). However, in our analyses, we have observed stronger effect of HDAC5 in overall deacetylation of AF9 (Fig. 7C, compare lane 4 vs lane 6). Therefore, for subsequent studies for role of deacetylation in controlling AF9-mediated functional regulation, we have focused on HDAC5. Consistent with a role of HDAC5 in regulating AF9 functions, we have also observed interaction between ectopically expressed HDAC5 and endogenous AF9 by our

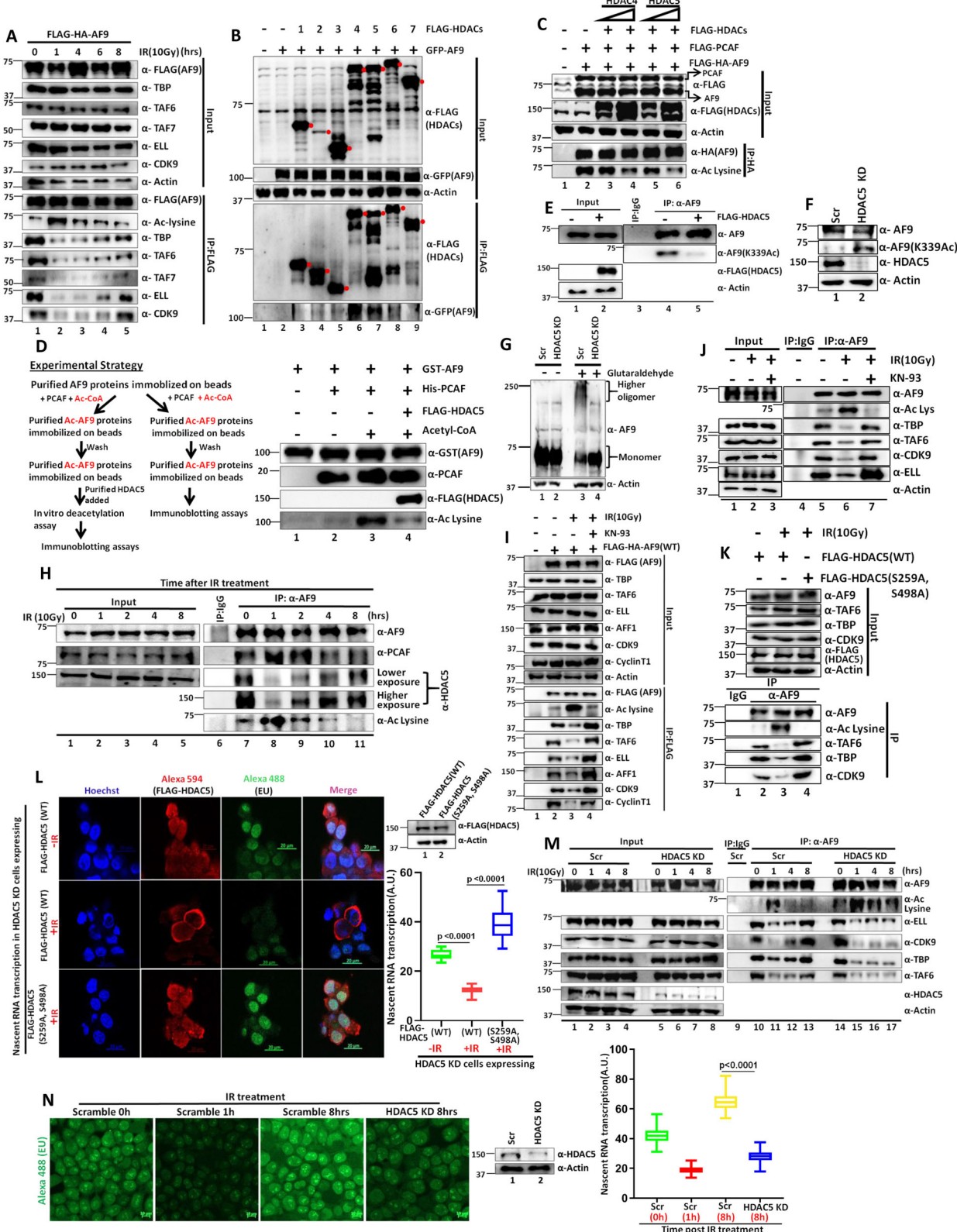

immunoprecipitation analysis (Supplementary Fig. 7A). To address a direct role of HDAC5 in deacetylating AF9, we purified full-length HDAC5 through its overexpression within mammalian cells and subsequent purification in high salt buffer (Supplementary Fig. 7B). Subsequent in vitro deacetylation assay using experimental strategy as shown in the left panel of Fig. 7D, clearly showed a direct role of HDAC5 in deacetylating AF9 in vitro (Fig. 7D, right panel, compare lane 3 vs

lane 4). Consistent with a role of HDAC5 in deacetylating AF9, its overexpression reduced acetylation of endogenous AF9 at K339 residue (Fig. 7E), whereas, knockdown of HDAC5 enhanced this acetylation (Fig. 7F). Thus, based on all these evidences, we conclude that human HDAC5 plays a key role in deacetylating AF9 at K339 residue that may have important roles in IR-dependent global transcriptional regulation. Consistent with this hypothesis, we have observed

**Fig. 7 | CaMKII-mediated phosphorylation, regulating nuclear-cytoplasmic shuttling of HDAC5, is important for genotoxic stress-dependent modulation of AF9 acetylation for transcriptional regulation within 293 T cells. A** Dynamic regulation of ectopically expressed AF9 acetylation and its concomitant association with indicated TFIID and SEC components after IR(10 Gy) treatment (*n* = 3 replicates). **B** Immunoblotting analysis showing interaction of ectopically expressed GFP-AF9 with indicated FLAG-HDACs within mammalian cells (*n* = 2 replicates). **C** Immunoblotting analysis showing deacetylation of ectopically expressed AF9 by concomitant expression of HDAC4 and HDAC5 (*n* = 2 replicates). **D** Immunoblotting analysis showing deacetylation of AF9 by purified HDAC5 in vitro (*n* = 1 replicate). **E** Immunoblotting analysis showing deacetylation of endogenous AF9 at K339 residue by ectopically expressed HDAC5 within mammalian cells (*n* = 1 replicate). **F** Immunoblotting analysis showing enhanced acetylation of endogenous AF9 at K339 residue upon HDAC5 knockdown (*n* = 2 replicates). **G** Immunoblotting analysis showing reduced AF9 oligomerization potential upon HDAC5 knockdown within mammalian cells (*n* = 2 replicates). **H** Immunoblotting analysis showing dynamic interaction of PCAF and HDAC5 with endogenous AF9 upon IR treatment (*n* = 1 replicate). **I, J** Immunoblotting analysis

showing effect of prior treatment with CaMKII inhibitor (KN-93) on IR-induced acetylation of both ectopic ((**I**) *n* = 3 replicates) and endogenous AF9 (**J**) *n* = 2 replicates) and concomitant interaction with TFIID and SEC components within mammalian cells. **K.** Immunoblotting analysis showing effect of ectopic expression of CaMKII-mediated phosphorylation-defective HDAC5(S259A, S498A) mutant on IR treatment-mediated interaction of TFIID and SEC components with endogenous AF9 (*n* = 2 replicates). **L** Nascent RNA transcription analysis showing enhanced global transcriptional ability in cells expressing HDAC5(S259A, S498A) mutant when compared to HDAC5(WT) after IR(10 Gy) treatment. The boxes represent median and quartiles and value ranges of 25–75 and 10–90%. Upper and lower hinges extend to the largest and smallest datapoints (*n* = 30 cells). This experiment was done once. **M** Effect of HDAC5 KD on IR treatment-dependent dynamic interaction of endogenous AF9 with TFIID and SEC components at indicated time periods (*n* = 1 replicate). **N** Nascent RNA transcription analysis (through EU incorporation) showing effect of HDAC5 KD on global transcription restart at 8hrs after IR(10 Gy) treatment. The boxes represent median and quartiles and value ranges of 25–75 and 10–90% (*n* = 100 cells). This experiment was done once.

restoration of self-association between ectopically expressed FLAG- and GFP-tagged AF9 when co-expressed with HDAC5 in presence of PCAF within mammalian cells (Supplementary Fig. 7C, compare lane 4 vs lanes 5, 6). Further, with enhanced acetylation of AF9 upon HDAC5 knockdown (Fig. 7F), we have observed enhanced monomerization of endogenous AF9 (Fig. 7G, compare lane 3 vs lane 4) which potentially can have functional implications in transcription after IR treatment.

## Nuclear and cytoplasmic shuttling of HDAC5 upon IR treatment plays a key role in regulation of AF9 acetylation and concomitant target gene expression upon IR treatment

For addressing the detailed role of HDAC5 in regulation of IR-dependent acetylation of AF9, our initial analyses showed reduced interaction with HDAC5 and concomitant enhanced interaction with PCAF with ectopically expressed AF9 at early time points (1 hr and 2hrs) after IR treatment (Supplementary Fig. 7D, compare lane 1 vs lanes 2, 3). However, at later time points (4 hrs and 8 hrs), we observed opposite interaction patterns, wherein, enhanced interaction with HDAC5 and reduced interaction with PCAF is observed (Supplementary Fig. 7D, compare lanes 2, 3 vs lanes 4, 5). Similar results are also observed when endogenous AF9 is immunoprecipitated at different time points after IR treatment (Fig. 7H). Further, consistent with enhanced PCAF along with reduced HDAC5 association, we also observed enhanced acetylation of endogenous AF9 at early time points (1 hr and 2hrs) after IR treatment (compare lane 7 vs lanes 8, 9). However, at subsequent time points, the acetylation level of AF9 was reduced (lanes 10,11).

What are the underlying mechanisms of differential HDAC5 interaction with AF9 at different time points after IR treatment? Both HDAC4 and HDAC5 belong to class II histone deacetylases[27,32]. Upon exposure to genotoxic stress, increased reactive oxygen species (ROS) generation leads to activation of Calcium/Calmodulin-dependent protein kinase II (CaMKII)[33,34]. The activated CaMKII, in turn, phosphorylates class II deacetylases including HDAC5 at two key residues (Ser259 and Ser498) leading to its nuclear export[35,36]. Indeed, upon IR treatment, we also observed shuttling out of endogenous HDAC5 from nucleus to cytoplasm at early 1 hr and 2hrs time points (Supplementary Fig. 7E, compare HDAC5 blots at 1 hr and 2hrs after IR treatment in both nuclear and cytoplasmic fractions). However, at later time points (at 4hrs and 8hrs), cytoplasmic HDAC5 shuttles back to nucleus. Interestingly, these are the time points when modulation of AF9 interaction with HDAC5 and PCAF is achieved (Supplementary Fig. 7D and H). Treatment of CaMKII-specific inhibitor (KN-93) impairs the shuttling of nuclear HDAC5 to cytoplasm at early time points after IR treatment (Supplementary Fig. 7F, compare lanes 2, 3 vs 7, 8 and 4, 5 vs 9, 10). The nuclear export of HDAC5 is important for concomitant

enhanced acetylation since, treatment of cells with KN-93 abrogated this enhanced acetylation of ectopically expressed AF9 (Fig. 7I, compare lane 3 vs lane 4). Further, consistent with a role of acetylation in reducing its interaction with TFIID, we also failed to observe reduced TFIID interaction with AF9 upon KN-93 treatment (Fig. 7I, compare lane 3 vs lane 4). Since prior AF9 acetylation is important for downstream DNA-PKc-dependent phosphorylation of AF9 and its subsequent reduced interaction, we also failed to observe decreased SEC interaction in cells pretreated with KN-93 followed by IR treatment (lane 3 vs lane 4). Similar results are also observed when endogenous AF9 is immunoprecipitated by using specific antibody (Fig. 7J, compare lane 6 vs lane 7). Consistent with all these observations, prior KN-93 treatment also failed to show efficient transcriptional downregulation upon IR treatment as assessed by qRT-PCR analysis of AF9-target genes (Supplementary Fig. 7G).

Next, we used CaMKII phosphorylation-defective HDAC5 mutant (S259A, S498A) that fails to show export of HDAC5 upon IR treatment when compared to the WT (Supplementary Fig. 7H, compare lanes 2, 3 vs 5, 6). Ectopic overexpression of the phosphorylation-defective HDAC5 mutant (S259A, S498A) showed defective AF9 acetylation and failed to show impaired TFIID and SEC association at 1 hr after IR treatment which otherwise is observed for HDAC5(WT) protein (Fig. 7K, compare lane 3 vs lane 4). Consistent with this observation, re-expression of the HDAC5 mutant (S259A, S498A) in HDAC5 KD cells also failed to show efficient global transcriptional downregulation by EU incorporation assay when compared to HDAC5(WT) (Fig. 7L, left panel for representative image and right panel for quantification). Similar results are also observed when individual AF9-target genes were assessed for their mRNA expression at 2hrs after IR treatment by qRT-PCR analysis (Supplementary Fig. 7I).

Thus, based on all these experiments, it is evident that, upon IR treatment, CaMKII-mediated phosphorylation of HDAC5 and its concomitant nuclear export causes cascade of reactions that involves, (a) PCAF-mediated acetylation of AF9 at K339 residue resulting in enhanced monomerization and reduced TFIID interaction, (b) AF9 K339 acetylation-dependent recruitment of DNA-PKc that leads to phosphorylation of AF9 at S395 residue resulting in reduced SEC interaction, (c) DNA-PKc-mediated recruitment of Ku complex for repair of damaged DNA by NHEJ pathway. Failure to export HDAC5 out of nucleus fails to show enhanced AF9 acetylation and resultant global transcriptional downregulation upon IR treatment.

From our analyses, it suggests that nuclear import of HDAC5 at later time points (4hrs and 8hrs) after initial export (at early time points) is important for deacetylating AF9 for restoring its interaction with cognate interacting partners for restarting transcription. Towards addressing this hypothesis, we used HDAC5 KD cells and monitored

acetylation dynamics of endogenous AF9 at different time points after IR treatment. Indeed, knockdown of HDAC5 resulted in persistence of acetylation of endogenous AF9 for longer time when compared to control scramble cells (Fig. 7M, compare lanes 10–13 vs lanes 14–17). Further, we also observed reduced AF9 interaction with other cognate interactors upon its acetylation and this reduced interaction was never restored (Fig. 7M, compare lane 14 vs lanes 15–17). However, parallel analysis in scramble cells showed full restoration of interaction of AF9 with other interacting partners (Fig. 7M, compare lane 10 vs lanes 11–13). Also, nascent RNA transcription analyses showed full restoration of transcription at 8hrs time point in control Scramble cells, whereas, the HDAC5 knockdown cells failed to do so (Fig. 7N, left panel for representative image and right panel for quantification). Consistent with this observation, mRNA expression analysis of AF9-target genes failed to show optimal transcriptional restart at 8hrs time point in the HDAC5 KD cells when compared to control Scr cells (Supplementary Fig. 7J). Thus, re-entry of HDAC5 plays a key role in regulation of restart of transcription after initial global downregulation through regulation of, at least in part, AF9 acetylation and corresponding restoration of factor interaction after repair of damaged DNA within mammalian cells.

### AF9 family protein ENL also shows similar genotoxic stress-dependent response

Since another AF9 family protein, ENL, also share similar property as that of AF9 in its interaction and functional regulation, we wondered whether functions of ENL would also be regulated through the mechanisms as deciphered by our analysis for AF9. Our initial analysis showed that, like AF9, ENL is also subjected to PCAF-mediated acetylation within mammalian cells (Supplementary Fig. 8A, compare lane 2 vs lane 3). Further, we have also observed IR treatment-dependent enhanced acetylation of ENL (Supplementary Fig. 8B, compare lane 1 vs lanes 2–4). However, unlike AF9, we observe prolonged ENL acetylation persistence after IR treatment such that even after 8hrs, we observe significant ENL acetylation. Consistent with the role of AF9 acetylation in reduced interaction with other interacting proteins, we also observed reduced interaction of TFIID and SEC with ENL upon IR treatment (Supplementary Fig. 8C, compare lane 2 vs lane 3). Similar results are also observed when we immunoprecipitated endogenous ENL using specific antibody (Supplementary Fig. 8D, compare lane 2 vs lane 3). Further, like AF9, co-expression of HDAC5 also reduces concomitant PCAF-mediated acetylation of ENL within mammalian cells suggesting a role for HDAC5 in deacetylation of ENL as well (Supplementary Fig. 8E, compare lane 3 vs lanes 4, 5). Thus, based on these data, we conclude that, like that of AF9, human ENL is also likely subjected through similar mechanisms for functional regulation during genotoxic stress. Further, stable knockdown of ENL cells also showed reduced growth potential like that of AF9 knockdown cells (Supplementary Fig. 8F G). Consistent with reduced growth potential, ENL knockdown also shows reduced expression of AF9-target genes that predominantly regulate proliferation of cells (Supplementary Fig. 8H). Interestingly, the AF9 knockdown cells did not show much effect on expression of ENL within mammalian cells (Supplementary Fig. 8I). This result thus suggests that the ENL protein, despite showing similar functional redundancies, does not functionally complement absence of AF9 protein. These observations also raise the possibility that both AF9 and ENL may regulate expression of same set of genes with their functional roles in regulation of different steps of transcription.

Since all of our above-mentioned experiments were performed in 293 T cells, we also addressed the cell-type specificity of overall regulation of AF9 functions upon IR treatment by using the HeLa cells. Like the one observed within 293 T cells, as shown in Supplementary Fig. 8J, we also observed similar response of endogenous AF9 acetylation upon IR treatment within HeLa cells as well, wherein, at early time point of 1 hr after IR treatment, we observed maximum acetylation of endogenous AF9 and is reduced afterwards at longer time points (4hrs and 8hrs time points). Further, consistent with a role of acetylation-dependent phosphorylation and concomitant regulation of interaction with other interacting partners, we have also observed enhanced acetylation and coupled DNA-PKc-mediated phosphorylation at early time point (1 hr) after IR treatment (Supplementary Fig. 8K, compare lane 7 vs lane 8). The enhanced acetylation and phosphorylation resulted in reduced interaction of AF9 with its interacting partners of TFIID and SEC components within HeLa cells as well. Thus, the overall mechanism, as described in this study, appears to be cell-type independent as well.

## Discussion

In this study, using human AF9 as a representative of elongation factor, we have deciphered comprehensive mechanistic insights into the dynamic regulation of events of transcription and DNA repair during genotoxic stress that involves two different post-translational modifications and associated players that are involved in these overall processes. Our study deciphers a role of YEATS domain of human AF9 in recognition of TFIID complex that critically depends on the Poly-Ser domain-mediated oligomerization. Interestingly, the overall effect of this mechanism of regulation is restricted for TFIID-dependent transcriptional regulation and does not affect functions that involves C-terminal domain-dependent SEC interaction. Interestingly, mammalian cells have come up with two-step regulations involving two different mechanisms for achieving efficient global transcriptional downregulation. Upon exposure to genotoxic stress, CaMKII-dependent phosphorylation of class II deacetylase HDAC5 leads to its nuclear export that, in turn, causes simultaneous enhanced PCAF-mediated acetylation at key K339 residue. Under normal condition, competitive functions of these two factors keep the AF9 K339 acetylation under check. Upon enhanced K339 acetylation, the AF9 protein becomes monomer and thus loses its ability to interact with TFIID and consequent transcriptional activation leading to global transcriptional repression. However, the acetylated AF9 still retains the ability to interact with SEC component with full potential. Therefore, further mechanism exists in which, the acetylated AF9 at K339 enhances its interaction with DNA-PKc. This interaction with DNA-PKc causes phosphorylation of AF9 at the downstream single S395 residue that results in loss of interaction with SEC components and further transcriptional repression. Since AF9 remains associated with chromatin, its enhanced interaction also helps in enhanced recruitment of DNA-PKc onto chromatin for recruiting downstream DNA repair factors such as Ku complex that helps in repair of damaged DNA. Once the damaged DNA is repaired, the HDAC5 deacetylase enters into nucleus and causes deacetylation and restoration of AF9 interaction with TFIID complex as well as SEC components for transcriptional restart. Thus, this study provides a comprehensive understanding of functional regulation of a key elongation factor for achieving global transcriptional repression, coupled with efficient repair as well as restart of transcription. The overall model for this mechanism of action is presented in Supplementary Fig. 9.

### Reversible oligomerization switch in regulation of global transcription

Oligomerization of proteins play important roles in differential association with cognate interactors and thus regulate physiological functions. In some cases, protein oligomerization is important for functional association, whereas, in other cases, reversible oligomerization states dictate the final physiological outcome in a context-dependent manner. For example, our earlier report has shown that the human ZMYND8 protein can remain either as monomer or dimer within mammalian cells. The monomeric ZMYND8 preferentially associates with the NuRD complex, whereas, the dimeric ZMYND8

associates with activator P-TEFb complex[23]. In this report, we have shown a function of human AF9 YEATS domain in interacting with TFIID complex for transcriptional activation. This interaction is critically dependent on the oligomerization property of the Poly-Ser domain. The sole function of Poly-Ser domain is to provide oligomerization property since replacing this domain with oligomerization domain, from another protein, fully restores the functional capacity to activate transcription. However, it has to be noted that these mechanisms of functional regulation are applicable only for AF9-target genes since the genes that are not the AF9 targets (Fig. 1E, F), would not likely show this mechanism of regulation for their expression within mammalian cells. Interestingly, this very same property is being elegantly used by mammalian cells to regulate the expression of genes in a context-dependent manner in which a single post-translational acetylation changes the oligomerization property and thus switches off the global transcription. This overall effect is reversible since, after the repair of damaged DNA, deacetylation by HDAC5 restores the cognate interactions and helps in restarting transcription. Thus, reversible switching between oligomer and monomer in a post-translational modification-dependent manner dictates the transcriptional response. Thus, this mechanism of action, as deciphered in this study, could pave the way for mechanistic understanding of context-dependent transcriptional regulation by several transcription factors that show propensity towards forming multimeric species through their self-association.

Since oligomerization of proteins is a key feature of liquid-liquid phase separation-dependent functional regulation within biological systems, it could highly be possible that similar mechanisms of functional regulation may also exist for AF9 protein as well. Consistent with this hypothesis, role of phase separation property of ENL and CyclinT1 in functional regulation of SEC has been described in few recent studies[37–39]. Further, the Poly-Ser domain is a classic example of intrinsically disordered region. Thus, functional regulation of AF9 through regulation of liquid-liquid phase transition is an interesting possibility that needs further exploration through future studies.

### Bi-functional regulation of a transcription factor involving two different post-translational modifications

Our earlier report has shown the presence of two distinct domains within AF9 protein that interacts with proteins involved in two different steps of transcription[18]. The C-terminal domain is involved in its interaction with SEC components, whereas, the N-terminus domain contains YEATS domain that interacts with acetylated and crotonylated histone[13,15], PAF1C[16], as well as TFIID. Further, apart from its interaction with TFIID, none of the other reported interactions require the Poly-Ser domain and thus its oligomerization capacity. Interestingly, AlphaFold-based structural prediction showed presence of structured regions restricted only within the C- and N-terminus region (Supplementary Fig. 2A). The remaining part of this protein forms the long-stretched loop region. From our study, it can be envisaged that post-translational modifications within the loop structures dictate the differential interactions by the structured interacting regions. For the AF9 protein, PCAF-mediated acetylation targets Poly-Ser domain-mediated oligomerization and thus the YEATS domain-mediated TFIID interaction for controlling the transcription through its interaction with the initiation factors. However, DNA-PKc-mediated phosphorylation of a single target residue controls its association with the other elongation components within SEC. Thus, two different mechanisms have been evolved to regulate both the steps for the AF9 protein which has bi-functional role in bridging the initiation and elongation steps of transcription through its two different domains. This similar mechanism of action in regulation of other bi-functional proteins can also be conceived for physiological response in a context-dependent manner.

### Acetylation-dependent phosphorylation modification in functional regulation of transcription

It is interesting to note that the overall transcriptional regulation involving AF9 during genotoxic stress is critically dependent on the initial acetylation modification. This is, in part, due to initial shuttling out of nucleus of AF9-specific deacetylase HDAC5. This causes concomitant enhanced acetylation of AF9 by PCAF at K339 residue resulting in its reduced interaction with TFIID components. The resultant acetylated AF9 performs another function of recruiting DNA-PKc onto chromatin through its enhanced interaction. This enhanced interaction further results in downstream phosphorylation at key S395 residue for abrogating SEC interaction. The reverse effect of phosphorylation-dependent acetylation does not hold true in this case since AF9 protein containing the phosphorylation-defective S395A mutant shows full acetylation capability at the target site (Fig. 5I). The acetylation-dependent phosphorylation mechanism in efficient downregulation of transcription during exposure to genotoxic stress is a unique mechanism wherein regulation of both the functions are important for achieving global transcriptional downregulation especially in the context of bi-functional AF9 protein that interacts with factors involved in initiation as well as elongation steps for overall functional regulation.

### Shuttling of class II deacetylases as key mechanisms in temporal regulations of functions of transcription as well as repair factors

Class II histone deacetylases possess unique properties of their abilities to shuttle in and out of nucleus in a post-translational modification-dependent manner. Upon exposure to genotoxic stress, increased level of reactive oxygen species (ROS) causes activation of CamKII that, in turn, phosphorylates the class II deacetylases including HDAC4 and HDAC5. The phosphorylated HDAC5 is shuttled out of nucleus and this causes an elevated level of acetylation of all the HDAC5 targets within the nucleus. AF9 being a target for HDAC5-mediated deacetylation, also responds through enhanced acetylation by PCAF. This enhanced acetylation, in turn, performs two key functions a) downregulates global transcription and b) helps in recruiting DNA repair factor DNA-PKc. The genotoxic stress-dependent shuttling out of class II deacetylases have implications in regulation of global transcription as well as repair of damaged DNA, which is essential for maintaining genomic integrity.

### Evolutionarily conserved mechanisms for dynamic regulations of functions for maintenance of genomic integrity and cell survival

Using AF9 as model transcription factor, our study has deciphered a mechanism of coordinated functional regulation of both transcription as well as DNA repair processes that involves post-translational acetylation and phosphorylation modifications by cognate acetyl transferase, kinase as well as deacetylase. This mechanism depicts a perfect example of cellular adaptive, evolutionarily conserved response for its survival upon exposure to genotoxic stress. Although being studied in details of functions of AF9, similar mechanism of action can also be envisaged by factors, which are also being targeted by class II deacetylases for their deacetylation and their corresponding response during genotoxic stress for functional regulation. For example, an earlier study showed that genotoxic stress-dependent shuttling out of HDAC5 promotes acetylation of p53 at K120 residue that in turn helps in expression of proapoptotic as well as apoptotic genes upon prolonged exposure to genotoxic stress[40].

### Implication of this study in mechanistic understanding of associated diseases

It is interesting to note that in majority of MLL-AF9 leukemic fusion cases, the C-terminal ends of AF9 are fused with the N-terminal end of MLL protein[7]. These C-terminal ends lack the key K339 and S395

residues that have shown to play important roles in regulation of transcription and repair of damaged DNA by our study as shown here. Based on these mechanistic understandings, it can be predicted that the presence of MLL-AF9 fusion proteins in the leukemic cells may interfere with the overall transcription as well as repair of damaged DNA upon exposure to genotoxic stress[41]. This could lead to a condition, wherein the MLL-AF9 harboring leukemic cells would rely excessively on DNA repair machineries for their survival and functions. Indeed, consistent with this hypothesis, a recent study has shown that combinatorial treatment of DNA damaging agent along with inhibitors of DNA repair pathways causes extreme sensitivity. Similar mechanisms of therapeutic implications can be envisaged using specific inhibitors of YEATS domain[42] in combination with DNA damaging agents for the purpose of controlling the excessive growth and proliferation of cancerous cells as well. Further future studies would be needed for better understanding of these therapeutic approaches.

## Methods

### Cell culture and transfection
Both HEK293T(293 T) and HeLa cell lines used in this study, were cultured in DMEM media (Gibco, USA) containing 10% FBS (Gibco, USA) and 1% penicillin–streptomycin (Invitrogen)at 37 °C, in presence of 5%CO$_2$. Transfection experiments using 293T cells were done using Fugene transfection reagent as per manufacturer's protocol. For immunoprecipitation or RNA analysis, cells were generally harvested at 48 hours post transfection, unless otherwise mentioned.

### Generation of plasmid constructs
All the plasmids were cloned into appropriate expression vectors as mentioned in Supplementary Data 1. For the purpose of target factor expression in mammalian cells, the constructs were cloned in pcDNA5-FRT-TO vector with respective epitope tags as mentioned. All the deletion constructs and point mutants of AF9 were cloned in pcDNA5-FRT-TO vector. For expression of factors tagged with GFP epitope, constructs were cloned into EGFP-pcDNA3 vector. To express proteins in the bacterial system, pET-11d and pET-GST vectors were used to clone His-tagged and GST-tagged constructs respectively. Suitable restriction enzymes were used for the purpose of each cloning. All the point mutants as well as the deletion mutants were generated through site-directed mutagenesis and were confirmed by sequencing before being used in experiments. The details of restriction enzymes and cloning methods would be available upon request.

### Generation of stable knockdown cells
For making stable knockdown, shRNAs targeting AF9, PCAF, HDAC5, and ENL were cloned into lentiviral pLKO.1-puro vector using the oligos as mentioned in Supplementary Data 4. The target shRNA constructs (500 ng) were co-transfected with 125 ng pMD2.G (envelope plasmid) and 375 ng pSPAX2 (packaging plasmid) in $3 \times 10^5$ cells in a single well of a 6-well plate. After 24–30 hrs, the old DMEM was changed with fresh complete DMEM. 72 hr post transfection, the lentiviral supernatant was collected and stored at −80 °C for subsequent use. Freshly seeded 293 T cells were transduced with 300 μl of respective lentiviral supernatant along with 8 μg/ml polybrene. 24 hrs post-transduction, cells were selected in presence of puromycin(3 μg/ml) containing media. The positive selected puromycin-resistant stably-integrated knockdown cells were subsequently checked for knockdown efficiency by western blotting using target factor-specific antibodies.

### Nuclear extract preparation
The cells were first harvested in 1× PBS and centrifuged at 800 × g for 5 min to measure the packed cell volume (PCV). The harvested cells were resuspended in 2 volumes of PCV of nuclear extraction 1 buffer (10 mM Tris-HCl pH 7.3, 10 mM NaCl, 1.5 mM MgCl$_2$, and 0.7 μl/ml β-mercaptoethanol). Cells were allowed to swell in hypotonic buffer in

ice for 15 mins. This was followed by resuspending and passaging of cells through 23-gauge syringe for 10 times. Then, the cell lysate was centrifuged at 3000 × g for 5 min at 4 °C. After discarding the supernatant carefully, the nuclear pellet volume (NPV) was estimated. The nuclear pellet was subsequently resuspended in 2 volumes of nuclear pellet volume (NPV) of pre-chilled nuclear extraction 2 buffer (20 mM Tris-Cl pH 7.3, 1.5 mM MgCl$_2$, 20 mM NaCl, 0.2 mM EDTA and 25% glycerol) containing protease inhibitor cocktail (Roche) and 0.7 μl/ml β-mercaptoethanol. This was followed by slow addition of 1 volume of NPV of pre-chilled nuclear extraction 3 buffer (20 mM Tris-Cl pH 7.3, 1.5 mM MgCl$_2$, 1.2 M NaCl, 0.2 mM EDTA, 25% glycerol) containing protease inhibitor cocktail. For efficient extraction, the lysate was kept on ice for 45–60 mins with intermittent vortexing after every 3 mins. The sample was centrifuged at 11,600 × g for 20 min at 4 °C. The supernatant was collected as nuclear extract and used for experimental analysis.

### Immunoprecipitation and western blot analysis
Respective epitope-tagged proteins were ectopically expressed by transfection in 293 T cells. 48 hr post transfection, the cells were harvested, unless otherwise mentioned, and resuspended in lysis buffer (20 mM Tris-Cl pH 8, 20% glycerol, 2 mM EDTA, 300 mM KCl), supplemented with 0.1% NP-40, protease inhibitor cocktail, 0.7 μl/ml β-mercaptoethanol and 2 mM PMSF. The lysates ectopically expressing FLAG- and HA-tagged proteins were subjected to immunoprecipitation by incubating with anti-FLAG(M2) agarose beads and anti-HA agarose beads respectively for 14hrs at 4 °C. The next day, after vigorous washing of the immunoprecipitated samples with the lysis buffer supplemented with 0.1% NP-40, the bound proteins were eluted by boiling at 95 °C for 8–10 min in 1× SDS-loading dye. The samples were run in 6–12% SDS-PAGE gel at 100 V for required period of time. The properly resolved proteins were then transferred onto nitrocellulose membrane in pre-chilled transfer buffer containing 10–15% methanol at 100 V for 2–3 hrs. The membranes were blocked with 5% skimmed milk (HiMedia) solution for 1 hr at room temperature. After washing to remove the excess blocking solution, the membranes were incubated with target primary antibodies with desired dilutions (as mentioned in Supplementary Data 2) for overnight at 4 °C. The following day, the antibody-bound membranes were washed for 3× with TBST (1× TBS + 0.1%Tween 20) followed by incubation with species-specific secondary antibodies. The membranes were again washed thrice with 1XTBST and blots were developed in Azure 300 gel imaging system (Azure Biosystems) or iBright imaging system (Thermo Fisher Scientific) using ECL (BioRad). Majority of our immunoprecipitation experiments involving both ectopically expressed and endogenous proteins represent at least $n = 2$ biological replicates.

### Immunoprecipitation of endogenous proteins
The 293 T cells were harvested in 1 × PBS, centrifuged at 800 × g at 4 °C for 5 min and the supernatant was discarded. The harvested pellet was lysed in BC300 buffer (20 mM Tris-Cl pH 8, 20% glycerol, 2 mM EDTA, 300 mM KCl), supplemented with 0.1% NP-40, protease inhibitor cocktail, 0.7 μl/ml β-mercaptoethanol and 2 mM PMSF. The lysed sample was spun at 11600 × g at 4 °C for 15 min. Subsequently, the whole cell lysate was initially pre-cleared with protein-G magnetic beads (Invitrogen) for 2 hr at 4 °C, and was used for IP. For all of our endogenous IP experiments, IgG was used as control. Simultaneously, protein-G magnetic beads were blocked with 1% BSA in BC300 buffer (supplemented with 0.1% NP-40) for 2hrs at 4 °C, followed by washing thrice with BC300 buffer (+0.1% NP-40) and incubating with 2 μg of target-specific antibodies and species-specific IgG antibody as well for control experiment. After pre-clearing, the cell lysates were subsequently incubated with antibody-bound protein-G magnetic beads for 12–14 hrs

at 4 °C. Then, the beads were washed three times with BC300 buffer (+0.1% NP-40) before eluting the bound proteins by boiling at 95 °C for 8–10 min in 1× SDS-loading dye for downstream western blot analysis.

### Recombinant protein purification

For purification of proteins from the bacterial systems, BL21(DE3) *E. coli* cells were transformed with respective plasmid constructs cloned in pET-GST vector (GST-AF9) or His-pET-11d vector (His-GFP-AF9 and His-PCAF HAT domain). For purification of His-tagged protein(His-GFP-AF9), protein expression was induced with 1 mM IPTG (GoldBio) and grown at 18 °C for 18 hr. Cells were harvested and resuspended in lysis buffer (50 mM Na$_2$HPO$_4$, 300 mM NaCl, 10 mM imidazole, 20% Glycerol) supplemented with 0.1% NP-40, protease inhibitor cocktail and 0.7 μl/ml β-mercaptoethanol. The sample was sonicated for 5 min (at 60% amplitude, with 30 sec pulses on and off) on ice and spun at 11600 × $g$ at 4 °C for 20 min. Ni-NTA beads was added to the supernatant and was allowed for binding for 4hrs at 4 °C. The bead-bound proteins were washed extensively in wash buffer (20 mM imidazole, 300 mM NaCl, 50 mM Na$_2$HPO$_4$, 20% glycerol pH 8; supplemented with 0.1% NP-40) and eluted in elution buffer (250 mM imidazole, 300 mM NaCl, 50 mM Na$_2$HPO$_4$, 20% glycerol pH 8; with 0.1% NP-40).

For purification of GST-AF9, protein expression was induced with 1 mM IPTG (GoldBio) and grown at 18 °C for 18hrs. The harvested cells were lysed by sonication in lysis buffer containing 2 mM EDTA, 20 mM Tris-Cl pH 8, 300 mM KCl, 20% glycerol; supplemented with protease inhibitor cocktail, 0.1% NP-40 and 0.7 μl/ml β-mercaptoethanol. The whole cell lysate was subsequently centrifuged at 11600 × $g$ for 20 min at 4 °C. The supernatant was incubated with glutathione agarose beads (Pierce) for 6–8 hrs at 4 °C. The bead-bound proteins were washed thoroughly and stored in buffer (20 mM Tris-Cl pH 8, 2 mM EDTA, 300 mM KCl, 20% glycerol) at 4 °C, for subsequent usage for in vitro interaction analysis assays. Otherwise, bead-bound GST-AF9 protein was washed thrice with BC300 buffer, and eluted in 100 mM Tris-Cl pH 8 containing 30mM L-Glutathione reduced (Sigma).

### Native PAGE analysis

Full-length His-GFP-AF9 proteins were purified from the bacterial expression system and mixed with 4× loading buffer (62.5 mM Tris-Cl pH 6.8, 25% glycerol and 0.01% bromophenol blue) without addition of reducing agent β-mercaptoethanol. The samples thus prepared for native PAGE analysis were run in non-denaturing gels (without SDS), at 4 °C using ice-cold Tris-glycine running buffer. Then, the desired bands were analyzed by Coomassie staining. 4 μg of BSA samples was prepared in a similar manner and was loaded onto the gel as molecular weight marker under native condition.

### In vitro cross-linking assay

The whole cell lysates prepared from knockdown (KD) cells (PCAF KD/ HDAC5 KD) or transfected 293 T cells were incubated with very mild concentration (0.02%) of cross-linking agent glutaraldehyde, and kept for cross-linking at room temperature for indicated time points. The ongoing cross-linking reactions were stopped by addition of 5× dye. This was followed by heating the samples at 95 °C for 10 min. The samples were then loaded onto SDS-PAGE followed by western blotting.

### In vitro interaction analyses

For studying direct in vitro interactions, the purified recombinant target proteins such as GST-AF9 or GST alone, were initially immobilized on glutathione agarose beads. In all these in vitro binding assays, purified GST protein was used as control. GST bead-bound proteins (GST alone, GST-AF9) were incubated with prey proteins (His-GFP-AF9) in binding buffer (20 mM Tris-Cl pH 8, 150 mM KCl, 20% glycerol, 2 mM EDTA) with BSA (20 ng/μl) and 0.1% NP-40, and kept for overnight binding at 4 °C. The beads were washed rigorously with binding buffer

(+ 0.1% NP-40), and subsequently eluted in 1× SDS-loading dye by heating at 95 °C for 8–10 min.

### Dot blot assay

2–4 μl of different concentrations of the peptides (modified and unmodified) was spotted very slowly onto the strip of nitrocellulose membrane, so that the solution penetrates the membrane minimally. After drying the membrane properly at room temperature for an hour, the membrane was blocked with 5% skimmed milk (HiMedia) for 1 hr at room temperature. After washing, the membranes were incubated with target primary antibody (AF9-K339Ac) with desired dilution for overnight at 4 °C. Next day, the antibody-bound membranes were washed for 3× with TBST (1× TBS + 0.1% Tween 20) followed by incubation with species-specific secondary antibodies. The membranes were again washed for 3× with 1× TBST and blots were developed in iBright imaging system (Thermo Fisher Scientific) using ECL (BioRad).

### In vitro acetylation assay

Full-length GST-AF9, purified from bacterial cells was used as substrate in the HAT assay. His-PCAF HAT domain was also purified from bacterial cells as mentioned earlier. Acetylation reactions were carried out by incubating purified GST-AF9 with purified His-PCAF HAT domain, at 30 °C for 2 hrs in acetylation buffer containing 75mMTris-Cl pH 8, 1.25 mM EDTA, 12.5 mM DTT, 0.25% Tween 20, 25% glycerol, in the presence or absence of acetyl-CoA (5 mM) with intermittent tapping. The reaction was stopped by adding SDS-loading dye to the acetylation reaction mixture, followed by boiling at 95 °C for 8–10 mins. Then, it was subjected to SDS-PAGE analysis and immunoblotting using pan-acetyl lysine-specific antibody.

For performing in vitro interaction analysis with acetylated GST-AF9 with purified TFIID complex, initially bead-bound GST-AF9 was acetylated with purified His-PCAF HAT domain following the above-mentioned protocol. After washing the acetylated bead-bound AF9, purified TFIID complex was added to the same and in vitro interaction was carried out in binding buffer (containing 20 mM Tris-Cl pH 8, 20% glycerol, 2 mM EDTA, 100 mM KCl) in presence of BSA (20 ng/μl) and 0.1% NP-40, at 4 °C.

### In vitro deacetylation assay

Purified bead-bound GST-AF9 protein was incubated with purified His-PCAF HAT domain and the acetylation reaction was carried out as mentioned earlier. Then, the bead-bound acetylated AF9 protein was washed thrice, and the supernatant was discarded. Subsequently, different concentrations of FLAG-HDAC5 protein purified from mammalian cells were added to the acetylated GST bead-bound AF9 in deacetylase buffer (50 mM Tris pH 8, 4 mM MgCl$_2$, 0.2 mM DTT) for 3 hr at 37 °C. The reaction was stopped by boiling with 1× SDS dye at 95 °C for 8–10 mins. Then, it was subjected to SDS-PAGE analysis and immunoblotting using pan-acetyl lysine-specific antibody.

### Human TFIID complex purification

Human TFIID complex purification was done following the same protocol as mentioned earlier[18]. For purification of human TFIID for in vitro studies, nuclear extract from 293 T cells expressing Flag-HA-tagged TBP protein was incubated with prewashed anti-FLAG agarose beads in a buffer containing 20 mM Tris pH 8, 20% glycerol, 2 mM EDTA, 150 mM KCl, 0.7 μl/ml β-mercaptoethanol, 2 mM PMSF and 0.1% NP-40, for overnight at 4 °C. After binding, the protein-bound beads were washed with binding buffer, and the bound proteins were eluted using elution buffer containing 3× Flag peptide (250 ng/ml) in 20 mM Tris pH 8, 20% glycerol, 2 mM EDTA, 100 mM KCl for 1 hr at 4 °C. The purified TFIID complex was used for in vitro interaction assays. Acetylated GST bead-bound AF9 was incubated with and without purified TFIID complex, and the in vitro interaction assay was carried out in binding buffer containing 20 mM Tris-Cl pH 8, 20% glycerol,

2 mM EDTA, 100 mM KCl in presence of BSA (20 ng/μl) and 0.1% NP-40, at 4 °C. The beads were washed extensively with binding buffer, containing 0.1% NP-40 and subsequently eluted in 1× SDS-loading dye by heating at 95 °C for 8–10 min and subjected to western blot analysis using factor-specific antibodies.

## Mass spectrometry analysis

Mass spectrometry analysis of AF9-associated proteins was done as described earlier[18]. Purified AF9.com was resolved on a 4–12% SDS-PAGE gel and proteins were stained using Coomassie Brilliant Blue-G. After careful excision, proteins were destained in 20% methanol for 6 hr. Reduced cysteines were alkylated using with IAA. Overnight digestion of proteins was done with trypsin. On a nanocapillary reverse phase column, the proteins were resolved witha 1% acetic acid/acetonitrile gradient, at a flow rate of 400 nl/min. After the peptides were separated, they were injected into anion-trap mass spectrometer (LTQ XL, Thermo Fisher). MS/MS spectra was determined and the proteins were detected by comparing against Human IPI database (v 3.41) using Tandem/Trans-Proteomic Pipeline (TPP) software.

## Chromatin fractionation

293 T cells were initially cross-linked with 1% formaldehyde (Sigma) for 10 min at room temperature. For IR-treated samples, the cells were exposed to IR(10 Gy) and after indicated time points, the cells were proceeded for the subsequent cross-linking step. After cross-linking, the reaction was stopped by addition of 125 mM glycine (Sigma) for 5 min at room temperature. The fixed cells were washed with 1× PBS, and the cells were scraped, collected in a tube, centrifuged at $800 \times g$ at 4 °C for 5 min. The harvested cells were resuspended in 2 volumes of PCV of nuclear extraction 1 buffer (10 mM Tris-HCl pH 7.3, 10 mM NaCl,1.5 mM MgCl$_2$ and 0.7 μl/ml β-mercaptoethanol). After proper swelling of cells in ice-cold hypotonic buffer, they were resuspended, passaged through 23-gauge syringe for 10 times, and centrifuged at $3000 \times g$ for 5 min at 4 °C. After discarding the supernatant carefully, the nuclear pellet volume (NPV) was estimated. The nuclear pellet was resuspended in 2 volumes of nuclear pellet volume (NPV) of pre-chilled nuclear extraction 2 buffer (20 mM Tris-Cl pH 7.3, 1.5 mM MgCl$_2$, 20 mM NaCl, 0.2 mM EDTA and 25% glycerol) containing protease inhibitor cocktail and 0.7 μl/ml β-mercaptoethanol. This was followed by addition of 1 volume of NPV of ice-cold nuclear extraction 3 buffer (20 mM Tris-Cl pH 7.3, 1.5 mM MgCl$_2$, 1.2 M NaCl, 0.2 mM EDTA, 25% glycerol) containing protease inhibitor cocktail. The lysate was kept on ice for 45–60 mins with intermittent vortexing for proper extraction. The sample was centrifuged at $11,600 \times g$ for 20 min at 4 °C. The supernatant was discarded and the pellet was resuspended in BC300 buffer supplemented with protease inhibitor, and sonicated using Bioruptor™ UCD-200 (Diagenode) sonicator for 25 min (30 sec pulses on and off). Then the sonicated samples were centrifuged at $11,600 \times g$ at 4 °C for 20 min and the cleared supernatant(chromatin fraction) was subjected to western blotting using indicated antibodies.

## Immunofluorescence assay

In 12-well plates or 35 mm dishes, 293 T cells were grown on coverslips. Transfection was done with respective epitope-tagged plasmid constructs. 48hrs post transfection, the cells were fixed in 4% paraformaldehyde (Sigma) for 15 min at room temperature. After washing the cells with 1× PBS once, the cells were permeabilized with 0.5% Triton-X for 15 min at room temperature before blocking the cells in 1% BSA (Sigma) for 1 hr at room temperature. Subsequently, the cells were incubated with respective primary antibodies (1:1000 dilutions) for 12 hr at 4 °C. This was followed by washing the cells with 1× PBS and incubating with species-specific secondary Alexa-fluor (594) antibodies (1:500 dilutions) at room temperature for 1 hr. After washing the cells again with 1× PBS, staining of nuclear DNA was done with

Hoechst dye for 15 min at room temperature. The cells were again washed and imaging was done with LSM 800 (ZEISS) confocal microscope and analyzed using Zen 2.3 lite software. For immunofluorescence experiments of endogenous proteins, the 293 T cells grown on coverslips in 12-well plates were fixed and proceeded for slide preparation following the above procedure.

## Nascent RNA transcription analysis

Click-iT™ RNA Alexa-Fluor™ 488 Imaging Kit (Invitrogen) was used to perform nascent RNA transcription analyses. 293 T cells were grown on coverslips in 12-well plates or 35 mm dishes and transfected with plasmid constructs expressing FLAG-AF9 (WT), FLAG-AF9(K339R), FLAG-AF9(S395A), FLAG-HDAC5(WT), FLAG-HDAC5(S259,498 A) respectively. 48hrs post transfection, the cells were treated once with IR(10 Gy). For KD-related experiments, control Scramble(Scr) and stable KD cells (PCAF/HDAC5) were seeded and treated once with IR(10 Gy). 15 min prior to completion of 1 hr of IR treatment, the cells were treated with 0.3 mM 5-ethynyl uridine (EU). Similar to immunofluorescence analysis procedure as mentioned above, the cells were fixed in 4% paraformaldehyde for 15 min, permeabilized in 0.5% Triton-X for 15 min and blocked in 1% BSA for 1 hr at room temperature. The permeabilized cells were incubated in Click-iT reaction cocktail for 30 min at room temperature, washed with rinse buffer at room temperature and incubated with primary antibody specific for FLAG-epitope tag (1:1000) overnight at 4 °C. Subsequently, the cells were washed with 1× PBS and incubated with species-specific secondary antibody Alexa-fluor 594 (1:500) at room temperature for 1 hr. After washing the cells again with 1× PBS, staining of nuclear DNA was done with Hoechst dye for 15 min at room temperature. The cells were further washed and imaging was done with LSM 800 (ZEISS) confocal microscope and analyzed using Zen 2.3 lite software.

## Fractionation of cytoplasmic and nuclear proteins

For studying cytoplasmic and nuclear localization of endogenous HDAC5 upon IR treatment, 293 T cells were treated with IR(10 Gy) and were harvested at the mentioned time points post IR treatment. For other studies with ectopically expressed proteins, 293 T cells were transfected with FLAG-HDAC5 (WT or S259A, 498 A) and subjected to IR(10 Gy) at 48 hrs post transfection. The cells were then harvested with 1× PBS and centrifuged at $800 \times g$ for 5 min to estimate the packed cell volume. The cell pellet was resuspended in $2 \times$ PCV of NE1 buffer (10 mM Tris-Cl pH 7.3,1.5 mM MgCl$_2$,10 mM NaCl) supplemented with protease inhibitor cocktail and 0.7 μl/ml β-mercaptoethanol and kept on ice for 20 mins. The cell suspension was passaged through a 23-gauge needle for 10 times.Then, it was centrifuged at $3000 \times g$ at 4 °C for 10 mins, to obtain the cytoplasmic fraction in the supernatant and the nuclear pellet. The cytoplasmic fraction was collected and further centrifuged at $11,600 \times g$ at 4 °C for 10 min, to eliminate any nuclear protein contamination. The nuclear pellet volume (NPV) obtained in the previous step was estimated and resuspended in 2xNPV of NE2 buffer (20 mM Tris-Cl pH 7.3, 1.5 mM MgCl$_2$, 20 mM NaCl, 0.2 mM EDTA and 25% glycerol) containing protease inhibitor cocktail and 0.7 μl/ml β-mercaptoethanol. 1xNPV of NE3 buffer (20 mM Tris-Cl pH 7.3,1.2 M NaCl, 1.5 mM MgCl$_2$, 0.2 mM EDTA, 25% glycerol) supplemented with protease inhibitor cocktail (Roche) and 0.7 μl/ml β-mercaptoethanolwas added and incubated on ice for 45 minutes. For efficient extraction of nuclear protein, the samples were vortexed intermittently on ice. To pellet the nuclear debris, the samples were centrifuged at $11,600 \times g$ for 20 minutes at 4 °C and the supernatant was collected as nuclear fraction. Thus, the nuclear and cytoplasmic fractions obtained were mixed with SDS-loading dye, boiled at 95 °C or 10 mins. The nuclear and cytoplasmic localization of the proteins were analyzed by western blotting, with appropriate nuclear and cytoplasmic markers.

### RNA extraction, reverse transcription, and qRT-PCR analyses

Total RNA was extracted from KD(AF9/HDAC5/ENL) or transfected 293 T cells of a 6-well plate, using TRIzol reagent (Invitrogen) following manufacturer's protocol. For RNA extraction after specific treatments, cells were treated with KN-93(10 μM) and subjected to IR(10 Gy) and cells were harvested after 2hrs, and proceeded for RNA extraction using TRIzol. 500 ng of RNA was reverse transcribed using verso cDNA synthesis kit (Thermo Scientific) using manufacturer's protocol. The synthesized cDNA was diluted 25× and used for subsequent qRT-PCR analysis. qRT-PCR was performed using target-specific primers as mentioned in Supplementary Data 3 and iTaq Universal SYBR Green Supermix (Biorad). The relative RNA analysis was done, after normalizing with expression of actin as internal control.

### Chromatin immunoprecipitation (ChIP) analysis

ChIP analysis was performed following the same protocol as mentioned earlier[23,43,44]. 293 T cells were transfected with the plasmids expressing AF9(WT) or various AF9-deletion constructs or mutants. 48hrs post transfection, the cells were initially cross-linked with 1% formaldehyde (Sigma) for 10 min at room temperature. For IR-treated samples, the cells were exposed to IR(10 Gy) and after 2hrs, the cells were proceeded for the subsequent cross-linking step. After cross-linking, the reaction was stopped by addition of 125 mM glycine (Sigma) for 5 min at room temperature. The fixed cells were washed with 1× PBS, and the cells were scraped, collected in a tube, centrifuged at $800 \times g$ at 4 °C for 5 min. The supernatant was discarded and the pellet was resuspended in ChIP lysis buffer (0.5% NP-40, 1% Triton-X-100, 150 mM NaCl, 20 mM Tris-Cl pH 7.5, 2 mM EDTA, protease inhibitor) and was kept on ice for 30 min. The ice-cold cell suspension was passaged through a 23-gauge syringe for 10 times and spun at $3000 \times g$ for 10 min at 4 °C The nuclear pellet was resuspended in ice-cold shearing buffer (1% SDS, 50 mM Tris-Cl pH 8, 10 mM EDTA) containing freshly added protease inhibitor cocktail (Roche) and sonicated using Bioruptor™ UCD-200 (Diagenode) sonicator for 25 min (30 sec pulse on and off). Then the sonicated samples were centrifuged at $11600 \times g$ at 4 °C for 20 min and the cleared supernatant was diluted 10× in ChIP dilution buffer (20 mM Tris-Cl pH 8, 1.1% Triton-X-100, 0.01% SDS, 1.1 mM EDTA and 167 mM NaCl). Initially, the diluted extracts were pre-cleared with IgG for 2hrs and then incubated with protein-G magnetic beads(Biorad)for 4hrs at 4 °C.The pre-cleared lysates were subjected to immunoprecipitation with 2 μg of desired target antibodies for 10–12hrs at 4 °C. Simultaneously, protein-G magnetic beads (Biorad) were blocked by incubating with dilution buffer containing salmon sperm DNA (4 μg/μl) for 10–12hrs at 4 °C. This was followed by addition of pre-blocked beads to the immunoprecipitated samples and were incubated further for 2hrs at 4 °C. The immunoprecipitated protein-G magnetic bead-bound proteins were washed sequentially. First, with low salt buffer (0.1% SDS, 1% Triton-X-100, 2 mM EDTA, 20 mM Tris-Cl pH 8, 150 mM NaCl; supplemented with protease inhibitor cocktail), second, with high salt buffer (0.1% SDS, 1% Triton-X-100, 2 mM EDTA, 20 mM Tris-Cl pH 8, 500 mM NaCl; supplemented with Protease inhibitor cocktail), third, with lithium chloride buffer (0.5 M LiCl, 1% NP-40, 1% deoxycholate, 20 mM Tris-Cl pH 8, 1 mM EDTA; supplemented with protease inhibitor cocktail) and finally with TE buffer (10 mM Tris-Cl pH 8, 1 mM EDTA). All the above washes were performed at 4 °C. The immunoprecipitated DNA was then eluted in elution buffer (1% SDS, 0.1 M NaHCO$_3$) at room temperature, with intermittent vortexing for efficient elution. The eluted DNA was reverse cross-linked by incubating with 200 mM NaCl at 65 °C overnight. The next day, the samples were further treated with Proteinase K (Sigma) at 65 °C for 1 hr. Immunoprecipitated DNA was subjected to purification using QIAquick PCR purification kit (Qiagen). The purified DNA was directly used as template for qRT-PCR analysis (Biorad CFX96™ Real-Time-System) using primers as mentioned in Supplementary Data 3 for analysis of enrichment of specific factors at indicated regions on target genes using specific primers. Details of all the chemicals that were used for all the methods as mentioned above and their sources are mentioned in Supplementary Data 5.

### Colony formation assay

AF9 KD cells were transfected with respective plasmids expressing empty vector (EV), AF9(WT), AF9 (Poly-SerΔ), AF9(Poly-SerΔ+p53 OD), AF9(Poly-SerΔ +p53 OD + YEATSΔ), AF9 (43–60Δ), AF9 (K339Q), and AF9(K339R) proteins. 48hrs after transfection, the cells were trypsinized, counted with a hemocytometer and ~$30 \times 10^4$ cells were seeded and allowed to grow for 7–10 days to form distinct colonies. The colonies were fixed with fixing solution consisting of methanol and acetic acid (3:1) for 15 min and subsequently stained in 0.5% crystal violet (in methanol) for 15 min at room temperature. The plates were subsequently washed thoroughly with water to remove excessive background stains and images were captured and comparative analysis of colony forming abilities was done. For comparative analysis of colony forming potential between cells ectopically expressing AF9(WT) and AF9(K339R) after IR treatment, AF9 KD cells transfected with these respective plasmids were exposed once to IR(10 Gy). 1 hr after IR exposure, the cells were trypsinized, counted with hemocytometer and ~$30 \times 10^4$ cells were seeded and grown for 7–10 days until formation of distinct colonies. The colonies were initially fixed, stained with crystal violet and washed to remove excess stain in a similar manner, followed by image capture.

### Cell proliferation assay

AF9 KD (293 T) cells were first transfected with respective plasmid constructs ((empty vector (EV), AF9(WT), AF9(43–60 aaΔ), AF9(K339Q), AF9(K339R)) as mentioned. 48hrs post transfection, ~$30 \times 10^4$ cells were seeded, and cells were counted by using hemocytometer on second and sixth day after seeding. For comparative analysis of cell proliferation ability between cells expressing AF9(WT) and AF9(K339R) after IR treatment, AF9 KD cells transfected with the respective plasmids were exposed to IR(10 Gy) at 48hrs after transfection. 1 hr post IR, the cells were trypsinized, counted with hemocytometer and ~$30 \times 10^4$ cells were seeded and counted on the mentioned days after seeding.

### Quantification and statistical analysis

All of our RNA expression and ChIP data represent means ± SD, from experiments of minimum two biological and three PCR replicates. Statistical analysis for our RNA and ChIP analyses were carried out using Graph Pad Prism (version 5.0) software. Statistical significance was calculated using one-tailed Student's $t$ test and $p$ values for each experimental diagram is mentioned on the top of the concerned figure.

### Reporting summary

Further information on research design is available in the Nature Portfolio Reporting Summary linked to this article.

## Data availability

The raw mass spectrometry spectra file is available via ProteomeXchange with the identifier PXD049385. The mass spectrometry analyses for identification of AF9-interacting proteins are available through public repository Mendeley database (https://data.mendeley.com/datasets/d2s63vgnmk/1). All original images for western blotting and microscopy analyses described in this study are provided as a Source data file. The raw datasets for ChIP, RNA, and nascent RNA transcription analyses are also provided in the Source file. Source data are provided with this paper.

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

## Acknowledgements

This work was supported by a DBT/Wellcome Trust India Alliance Senior Fellowship (IA/S/22/1/506227), and Council of Scientific and Industrial Research (CSIR) focused basic research (FBR) project MLP-141 awarded to D.B. and institutional support through P-07 budget head. We also greatly acknowledge Arijit Nandy for generating ENL expression constructs that were used in this study. Subham Basu is also acknowledged for generating GFP-AF9(WT) and GFP-AF9(Poly-SerΔ) expression constructs. P.T. is a recipient of the Indian Council of Medical Research (ICMR) Senior Research Fellowship ((SRF with fellowship ID number 3/1/3/JRF-2018/HRD-060(62039)). S.P. is a recipient of CSIR-SRF.

## Author contributions

P.T. performed majority of the experiments in consultation with D.B. S.P. performed nascent RNA transcription kinetics experiments in consultation with P.T. and D.B. P.T. and D.B. wrote the manuscript.

## Competing interests

The authors declare no competing interests.
