## [Peer Review File · Nature Communications]

Post-translational modification-dependent oligomerization switch in regulation of global transcription and DNA damage repair during genotoxic stressREVIEWER COMMENTS

Reviewer #1 (Remarks to the Author):

This work reports a novel mechanism of transcriptional regulation and DNA damage repair by the human AF9 protein, a component of the Super Elongation Complex (SEC). The authors show that AF9 interacts with TFIID through its Poly-Serine and YEATS domains, and this interaction is required for SEC recruitment and release of paused RNA polymerase II (Pol II) on target genes. The authors also show that upon exposure to genotoxic stress, AF9 undergoes PCAF-mediated acetylation at K339 residue, which causes monomerization of AF9 and reduces its interaction with TFIID, leading to global transcriptional repression and efficient DNA repair. The authors provide evidence from various experiments, such as deletion and mutation analyses, co-immunoprecipitation, chromatin immunoprecipitation, nascent RNA transcription, and colony formation assays, to support their findings.

The article is well-written and organized, and the data are clear and consistent. The authors use appropriate methods and controls to validate their results. The article contributes to the understanding of the molecular mechanisms of transcriptional regulation and DNA damage response in mammalian cells.

Specific comments

1. The authors revealed a comprehensive mechanistic insight into the dynamic regulation of events of transcription and DNA repair during genotoxic stress. However, they conducted most experiments by ectopic expression of tagged constructs and determining subsequent events. In many cases, these results are meaningful, while some molecular events of endogenous factors under physiological condition should be further addressed. For example, Fig. 3D showed that PCAF is responsible for the acetylation of AF9, while GCN5 and p300 are not involved. Could the authors provide the evidences of the specific interaction or colocalization of endogenous AF9 with PCAF, not with GCN5 or p300 in cells upon IR treatment? Similarly, could authors show specific interaction or colocalization of endogenous AF9 with DNA-PKc, not with ATR or ATM in Fig. 5?
2. According to the records on uniprot, the NLS is AAs 295-300 in AF9, AKKRKK. In your Supp. Figure 1, it seems you labeled AAs 383-391. Could you please provide more information to support that this fragment is an NLS, since SSSSSSSF seems different from any NLS sequences that I have ever seen?
3. Fig. S2B alone does not support the point of AF9 self-association. Any disruption of the self-association of AF9 should not lead to changes in this overlap unless the subcellular localization of AF9 is affected at the same time. Overlap of the signals from the two channels in this image at this resolution indicates aggregation of AF9, and any possible molecular interaction should be addressed using techniques like FRET, with which changes due to AF9 monomerization can be demonstrated. Do changes of AF9 oligomerization also affect its subcellular localization? Have the authors tried mass spectrometry or anything else to further confirm the oligomerization states AF9?

4. The authors could discuss the implications of their findings for the development of therapeutic strategies for diseases associated with aberrant transcription or DNA damage, such as cancer or neurodegeneration. They could also compare their results with previous studies on other SEC components or YEATS domain-containing proteins. Data from animal models could help readers understand this significance of the regulation demonstrated in this research, if it holds true in vivo.

Reviewer #2 (Remarks to the Author):

In this manuscript, the authors showed that genotoxic stress leads to PCAF-mediated acetylation of K339 residue of AF9 that causes monomerization of AF9 and reduces AF9-TFIID interaction for transcriptional downregulation. Furthermore, the authors showed that the K339 acetylation enhances AF9-DNA-PKc interaction which causes phosphorylation at S395 residue by DNA-PKc for reduced AF9-SEC interaction for further transcriptional downregulation. These modification of AF9 play a role in efficient repair of DNA damage. Finally, the authors showed that CaMKII-mediated phosphorylation-dependent nuclear export of HDAC5 enhances PCAF-mediated acetylation of K339 residue that causes monomerization of AF9 and reduces its interaction with TFIID and SEC for transcriptional downregulation. After repair, nuclear re-entry of HDAC5 reduces AF9 acetylation and restores its TFIID and SEC interaction for restart of transcription for survival of cells.

The authors showed interesting results. Especially, the manuscript includes many of biochemically results which are convincing. However, I think that several points should be addressed for publication in Nature Communications.

(1) The authors proposed that acetylation or phosphorylation of AF9 reduces oligomerized AF9 through the results of the manuscript. Is the oligomerization related to liquid-liquid phase separation of AF9? Considering that SEC has been shown that it forms liquid droplets in cells, this point should be addressed in the manuscript.

(2) The authors showed that acetylation and phosphorylation of AF9 reduced AF9 interaction with TFIID and SEC, respectively. In addition, the authors showed that replacement of endogenous AF9 with AF9 mutants in cells caused reduction of the occupancy of TBP and SEC at the promoters of AF9 target genes (Fig. 1I, 2L, 4H). These data raise the possibility that AF9 mutations which affect acetylation or phosphorylation of AF9 cause reduction of both transcription initiation and elongation. The authors should address whether AF9 mutation affect transcription initiation, elongation or both of them. This is important point to be addressed in this manuscript.

(3) Related to the question of (2), the authors proposed that AF9 mutants including AF9 (delta 43-60) or AF9 (Poly-Ser-delta) increased Pol II pausing (Fig. 1L, 2L). Is this really paused Pol II? How do the authors distinguish Pol II recruited to the promoters and Pol II paused at promoter proximal region?

(4) In contrast, the authors proposed that expression of AF9 K339Q which mimics acetylated AF9 promoted Pol II release (Fig. 4H). Is this true? In general, mNET-seq or PRO-seq are used to investigate Pol II pausing and release.

(5) In western blot data, the authors only showed ELL and CDK9 as SEC components. The authors should show at least one more SEC components, for example, AFF4, AF4 or ENL.

(6) In mass spec data, the authors showed only spectral counts of DNA-PKc. Did the mass spec analysis identify Ku70 and Ku80? The authors should show the proteins list identified in the mass spec analysis.

Reviewer #3 (Remarks to the Author):

In this paper, the authors use a deletion analysis coupled with IP's to show that the AF9 YEATS domain is essential for the interaction with TFIID, but not with components of the SEC such as CDK9 and ELL. They identify several genes that have reduced expression (and reduced SEC and TBP binding) upon AF9 KD, and show that YEATS 43-60aa deletions fail to rescue gene expression. They go on to show that a Poly-Serine domain is essential for AF9 self association and interaction with TFIID. Interestingly, a p53 oligomerization domain restores self association and TFIID interaction, and the rest of the YEATS domain contributes to the TFIID interaction but not the self association function, as measured by IP's, RT-PCR and ChIP.

To analyse AF9 in a more dynamic system, the authors treat 293T cells with ionizing radiation and show that AF9 displays reduced oligomerization, reduced TFIID and SEC interactions, and overall the cells display reduced transcriptional activity, and this is associated with increased acetylation of AF9. Acetylation of AF9 is controlled by PCAF on residue K339 (and deacetylated by HDAC5), and a K339Q acetyl mimic failed to rescue transcriptional defects in AF9 KD cells. With a series of detailed experiments they also show that DNA-PKc mediated phosphorylation at S395 reduces SEC binding to AF9, that a DNA-PKc interaction is dependent on K339 acetylation, and that components of NHEJ repair complexes are recruited to chromatin. The overall model is that acetylation of AF9 by PCAF reduces oligomerization and helps recruit DNA-PKc and disrupt SEC/TFIID binding to reduce transcription and enhance DNA repair upon irradiation.

This is an exciting, robust paper that builds on the author's previous work on AF9 function and significantly contributes to an important and expanding area of research. I have a few comments and questions, but overall this is excellent work.

Major questions

1. Do the AF9 KD cells grow over the long term? Do they have other defects other than reduced colony

forming capabilities?

2. As the authors indicate, ENL has a similar structure to AF9, and the proteins are considered to be possibly functionally interchangeable. Is ENL expressed in 293T cells? If so, do ENL knockdowns impact expression of similar genes as the AF9 targets in Figure 1?
3. Do ENL knockdowns also cause defects in colony formation of 293T cells?

Minor points and questions

1. Is there any evidence from the mass spec data for acetylation of AF9?
2. In the paper, the authors consistently refer to AF9 as having a role in global transcription regulation. From Figure 1E there appear to be genes that are insensitive to loss of AF9. A few words explaining this discrepancy should be made in the discussion.
3. Do other YEATS family (e.g. YEATS2, YEATS4) proteins contain a Poly-Ser stretch?
4. Although at one point the authors mention that all the work was done in 293T cells (except the final few experiments in HeLa cells), it would be helpful to indicate in the figure legends exactly what cells were used for each experiment.
5. The legends should also include n numbers for all the experiments. Are the many blots representative of multiple or single experiments?
6. The paper could overall use some editing for clarity.

RE: NCOMMS-23-44203-T-R1

Post-translational modification-dependent oligomerization switch in regulation of global transcription and DNA damage repair during genotoxic stress

Prathama Talukdar, Sujay Pal, and Debabrata Biswas

Authors' Response to Reviewers' Comments

We greatly appreciate the insightful comments and constructive feedback from all the reviewers that have immensely helped us to improve our manuscript. We have revised and extended our current manuscript with both modifications to the text and new experimental data according to the comments made by the Reviewers. Key experiments, amongst others, have been added to the revised manuscript as mentioned below.

A. As per the suggestion of Reviewer 1, we have performed our experiment to show PCAF-specific interaction with AF9 in presence of IR treatment. Along the same line, we have also shown DNAPKc-specific interaction with AF9 within mammalian cells.

B. Further, as per the suggestion of Reviewer 1, we have shown that the deletion of Poly-Ser domain of AF9 does not affect its subcellular localization thus ruling out any effect of impaired localization on oligomerization property of AF9.

C. We have also modified our text to incorporate some of the suggestions made by all the Reviewers for improving our manuscript.

D. As per the suggestion of Reviewer 2, we have performed the pausing index analyses in presence of several AF9 mutants (as indicated) to address the overall effect on pausing of Pol II at the promoter proximal region. Results of all these pausing index analyses have been incorporated into the main figures of the corresponding data section.

E. One of the key points raised by Reviewer 2 in relation to the role of liquid-liquid phase separation of AF9 in overall functional regulation is a very valid one. In fact, we also think along the same line for the overall regulation and are currently pursuing in that area. However, detailed analyses of these events and its overall implication is beyond the scope of this study.

F. Based on the suggestion by Reviewer 2, we have repeated our key interaction analyses to show the overall effect on AF9 interactions with AFF1 component of Super Elongation Complex (SEC) as well. These new figures have replaced the earlier ones in this revised manuscript.

G. As per the suggestion of Reviewer 3, we have addressed the overall effect of AF9 knockdown on expression of ENL protein within mammalian 293T cells.

H. Further, based on the suggestion by Reviewer 3, we have checked target gene expression and colony forming potential of 293T cells upon knockdown of ENL.

We are happy to note the extreme enthusiasm expressed by all the Reviewers towards our study deciphering mechanisms of functional regulation of AF9 through regulation of its oligomerization involving key post-translational modifications. Indeed, as pointed out by all the Reviewers, this study has the potential of significantly enhancing overall mechanistic understanding of dynamic regulation of global transcription and DNA damage repair within mammalian cells. We further hope that through the inputs, as suggested by all the Reviewers, this improved manuscript would now be ready to be accepted for publication in *Nature Communications*.

Authors' point-by-point response to Reviewers' Comments (02nd Nov 2023)

Reviewers' comments in blue

Authors' response in black

Reviewer #1 (Remarks to the Author):

This work reports a novel mechanism of transcriptional regulation and DNA damage repair by the human AF9 protein, a component of the Super Elongation Complex (SEC). The authors show that AF9 interacts with TFIID through its Poly-Serine and YEATS domains, and this interaction is required for SEC recruitment and release of paused RNA polymerase II (Pol II) on target genes. The authors also show that upon exposure to genotoxic stress, AF9 undergoes PCAF-mediated acetylation at K339 residue, which causes monomerization of AF9 and reduces its interaction with TFIID, leading to global transcriptional repression and efficient DNA repair. The authors provide evidence from various experiments, such as deletion and mutation analyses, co-immunoprecipitation, chromatin immunoprecipitation, nascent RNA transcription, and colony formation assays, to support their findings. The article is well-written and organized, and the data are clear and consistent. The authors use appropriate methods and controls to validate their results. The article contributes to the understanding of the molecular mechanisms of transcriptional regulation and DNA damage response in mammalian cells.

We are ecstatic seeing the positive comments from this Reviewer about suitability of our study for its publication in *Nature Communications*. We also would like to thank the Reviewer for all the constructive comments that have greatly helped in improving our manuscript.

Specific Comments:

1. The authors revealed a comprehensive mechanistic insight into the dynamic regulation of events of transcription and DNA repair during genotoxic stress. However, they conducted most experiments by ectopic expression of tagged constructs and determining subsequent events. In many cases, these results are meaningful, while some molecular events of endogenous factors under physiological condition should be further addressed.

We thank the Reviewer for this comment. However, with all due respect, we disagree with the Reviewer's comments that majority of our experiments are performed through ectopically-expressed proteins. We were very careful while designing our experiments especially keeping in mind the issue that the Reviewer has raised such that most of the experiments with ectopic expression have further been substantiated in the context of endogenous proteins as well.

For example, Fig. 3D showed that PCAF is responsible for the acetylation of AF9, while GCN5 and p300 are not involved. Could the authors provide the evidences of the specific interaction or colocalization of endogenous AF9 with PCAF, not with GCN5 or p300 in cells upon IR treatment?

To address the comment made by the Reviewer regarding PCAF-specific acetylation of AF9, we performed immunoprecipitation experiments of endogenous AF9 that clearly showed PCAF-specific interaction with AF9 upon IR treatment and not with co-expressed GCN5 or p300.

In this revised manuscript, we have incorporated this data as Fig. S3H.

Similarly, could authors show specific interaction or colocalization of endogenous AF9 with DNA-PKc, not with ATR or ATM in Fig. 5?

Similar to the above experiment, we have also performed immunoprecipitation of ectopic as well as endogenous AF9 that clearly showed stronger interaction of AF9 with that of DNAPKc than ATM in both the context.

In this revised manuscript, we have incorporated this data as Fig. S5B (for ectopic AF9) and S5C (for endogenous AF9).

2. According to the records on uniprot, the NLS is AAs 295-300 in AF9, AKKRKK. In your Supp. Figure 1, it seems you labeled AAs 383-391. Could you please provide more information to support that this fragment is an NLS, since SSSSSSSF seems different from any NLS sequences that I have ever seen?

We apologize for this inadvertent mistake from our end regarding this labeling. Indeed, it should be from 295-300 as the Reviewer has pointed out. We really appreciate the Reviewer for pointing this out. In this revised manuscript, we have corrected this mistake from our end.

3. Fig. S2B alone does not support the point of AF9 self-association. Any disruption of the self-association of AF9 should not lead to changes in this overlap unless the subcellular localization of AF9 is affected at the same time. Overlap of the signals from the two channels in this image at this resolution indicates aggregation of AF9, and any possible molecular interaction should be addressed using techniques like FRET, with which changes due to AF9 monomerization can be demonstrated. Do changes of AF9 oligomerization also affect its subcellular localization?

We thank the Reviewer for this very insightful comment. To address the comment on changes in subcellular localization of AF9 upon monomerization, we have checked localization of AF9(Poly-Ser Δ) that predominantly forms the monomer in our assays. As shown below, deletion of Poly-Ser domain resulting in monomerization of AF9, does not cause changes in its nuclear localization both by our microscopy assays as well as cellular fractionation and subsequent immunoblotting assay.

In this revised manuscript, we have incorporated this data as Fig.S2I (for microscopic assay) and S2J (for biochemical fractionation assay).

Have the authors tried mass spectrometry or anything else to further confirm the oligomerization states AF9?

No, we have not used mass spectrometry for confirming oligomerization. However, the other evidences as presented in this study convincingly confirm oligomerization of AF9 protein both *in vitro* as well as *in vivo* within mammalian cells.

4. The authors could discuss the implications of their findings for the development of therapeutic strategies for diseases associated with aberrant transcription or DNA damage, such as cancer or neurodegeneration.

These suggestions made by the Reviewer for improving our manuscript are really good and we would like to appreciate the Reviewer for making these constructive suggestions. Some of these aspects have been discussed in the "Discussion" section of this revised manuscript.

They could also compare their results with previous studies on other SEC components or YEATS domain-containing proteins. Data from animal models could help readers understand this significance of the regulation demonstrated in this research, if it holds true *in vivo*.

These are indeed very good suggestions by the Reviewer. However, all these detailed studies are beyond the scope of this study and would be followed up in subsequent studies.

Reviewer #2 (Remarks to the Author):

In this manuscript, the authors showed that genotoxic stress leads to PCAF-mediated acetylation of K339 residue of AF9 that causes monomerization of AF9 and reduces AF9-TFIID interaction for transcriptional downregulation. Furthermore, the authors showed that the K339 acetylation enhances AF9-DNA-PKc interaction which causes phosphorylation at S395 residue by DNA-PKc for reduced AF9-SEC interaction for further transcriptional downregulation. These modification of AF9 play a role in efficient repair of DNA damage. Finally, the authors showed that CaMKII-mediated phosphorylation-dependent nuclear export of HDAC5 enhances PCAF-mediated acetylation of K339 residue that causes monomerization of AF9 and reduces its interaction with TFIID and SEC for transcriptional downregulation. After repair, nuclear re-entry of HDAC5 reduces AF9 acetylation and restores its TFIID and SEC interaction for restart of transcription for survival of cells. The authors showed interesting results. Especially, the manuscript includes many of biochemically results which are convincing.

We really appreciate the Reviewer and are happy to receive his/her positive response regarding our submitted manuscript for its suitability for publication in *Nature Communications*.

However, I think that several points should be addressed for publication in *Nature Communications*.

We appreciate the constructive comments made by this Reviewer and have been happy to address all of them.

(1) The authors proposed that acetylation or phosphorylation of AF9 reduces oligomerized AF9 through the results of the manuscript. Is the oligomerization related to liquid-liquid phase separation of AF9? Considering that SEC has been shown that it forms liquid droplets in cells, this point should be addressed in the manuscript.

This could as well be a possibility considering that the Poly-Ser domain itself is a highly disordered region. Further, deletion of Poly-Ser domain reduces the puncta/condensate size of AF9 within the nucleus as shown in Fig. S2I of this revised manuscript. Also, consistent with a role of liquid-liquid phase separation in regulating its functions, we have also observed droplet formation by purified GFP-AF9 protein in our *in vitro* experimental setup as shown below.

Further, nuclear condensates formed by GFP-AF9 within mammalian cells are also sensitive to treatment of 1,6-Hexanediol that disrupts the hydrophobic interaction (shown below)

These results are shown just to address the comments made by this Reviewer. However, we do not wish to publish this data as part of this manuscript. This is a result of an ongoing study in our lab for detailed understanding of role of liquid-liquid phase transition of AF9 in regulation of transcription in a context-dependent manner. Thus, the overall role of liquid-liquid phase separation in functional regulation of AF9 is really an interesting question, however beyond the scope of this study.

(2) The authors showed that acetylation and phosphorylation of AF9 reduced AF9 interaction with TFIID and SEC, respectively. In addition, the authors showed that replacement of endogenous AF9 with AF9 mutants in cells caused reduction of the occupancy of TBP and SEC at the promoters of AF9 target genes (Fig. 1I, 2L, 4H). These data raise the possibility that AF9 mutations which affect acetylation or phosphorylation of AF9 cause reduction of both transcription initiation and elongation. The authors should address whether AF9 mutation affect transcription initiation, elongation or both of them. This is important point to be addressed in this manuscript.

We would like to thank the Reviewer for raising this constructive question. We have been happy to address this point by analyzing the pausing index (a ratio of Pol II being present at ~TSS/Coding region) in presence of AF9 mutants (within the AF9 knockdown cells). The pausing index analysis depicts the overall release of Pol II from the TSS region and thus its entry into the coding region. In all the genes that we have tested, we used ~3kb downstream of TSS region for analyzing the presence of amount of Pol II at the coding region.

(3) Related to the question of (2), the authors proposed that AF9 mutants including AF9 (delta 43-60) or AF9 (Poly-Ser-delta) increased Pol II pausing (Fig. 1L, 2L). Is this really paused Pol II?

With all due respect, we would like to point that the Reviewer may have intended to mention the Fig. 1I and not 1L (as suggested) since there was no figure 1L in our submitted manuscript. Therefore, we have addressed the comments for Fig. 1I and 2L. To address these comments, we have analyzed pausing index of Pol II as mentioned above in presence of these two AF9 mutants. The pausing index analysis on the target genes (as shown below) clearly shows that both these mutants enhanced Pol II pausing when compared to AF9(WT). While re-expression of AF9(WT) in the knockdown cells reduced the pausing (when compared to control EV), expression of these two proteins failed to do so on the target genes.

The effect of AF9 (43-60 aa) is shown in Fig. 1I and the effect of AF9 (Poly-SerΔ) is shown as part of pausing index analysis for Figure 2L along with other mutants as shown above. In all the cases where AF9 regained its interaction with TFIID, also showed reduced pausing on the target genes as shown above. However, mutants that showed reduced TFIID interaction, also showed enhanced Pol II pausing.

How do the authors distinguish Pol II recruited to the promoters and Pol II paused at promoter proximal region?

As such we can't distinguish between Pol II being recruited vs paused with our analyses as mentioned in this study. However, role of SEC components in assisting release the transcriptionally-engaged Pol II

have been demonstrated by multiple earlier studies. The presence of AF9 mutants that fail to interact with TFIID components, thus also fail to get recruited leading to impaired SEC recruitment at the target genes. This impaired recruitment further reduces release of Pol II that is being recruited at the promoter proximal region. Since SEC plays a major role in releasing paused Pol II, we conclude that its impaired recruitment in presence of AF9 mutants also causes a defect in release of paused Pol II. Moreover, our pausing index analysis (as shown in earlier comment), also shows significant effect of these AF9 mutants in increasing pausing of Pol II when compared to WT.

(4) In contrast, the authors proposed that expression of AF9 K339Q which mimics acetylated AF9 promoted Pol II release (Fig. 4H). Is this true? In general, mNET-seq or PRO-seq are used to investigate Pol II pausing and release.

With all due respect, we once again disagree with the Reviewer on this point. The AF9 K339Q mutant mimics the characteristics of acetylated AF9 and thus predominantly forms monomer and shows reduced TFIID interaction without losing SEC interaction. This reduced TFIID interaction further impairs

Figure R7: Pausing index analysis showing effect of indicated AF9 proteins on pausing of Pol II at the promoter proximal region of indicated genes as shown for Fig. 4H.

AF9-dependent SEC component recruitment on the target genes leading to enhanced Pol II pausing. Further, our pausing index analysis (as shown above) also supports the idea that AF9 K339Q mutant enhances Pol II pausing on the target genes when compared to AF9(WT).

Results of all the pausing index analyses are presented in the right side of the concerned figures in this revised manuscript. We assumed that this way of presentation would better integrate the overall data.

(5) In western blot datas, the authors only showed ELL and CDK9 as SEC components. The authors should show at least one more SEC components, for example, AFF4, AF4 or ENL.

In our submitted manuscript, we used two representative components of TFIID-interacting AF9•AFF1•P-TEFb and ELL•EAF1 complexes as shown in our last published study (Yadav et al., 2019). We strongly believe that representation by these components would sufficiently address overall functional regulation. However, to address the comment made by the Reviewer, we have performed

new experiments for some of the key figures (as shown above) and replaced them in this manuscript for above figures to show the effect with AFF1 component of SEC as well.

(6) In mass spec data, the authors showed only spectral counts of DNA-PKc. Did the mass spec analysis identify Ku70 and Ku80?

In the mass spectrometric analysis, we only observed presence of DNAPKc. We failed to observe any interaction with Ku70 and Ku80. The potential reason could as well be experimental condition in which the interacting proteins were immunoprecipitated. In case of mass spectrometry analysis, we employed 300mM KCl solution by following two-step tandem affinity purification for identifying strong interactors of AF9 in our experimental setup. However, for the purpose of immunoprecipitation analyses for showing interactions, we have used one-step FLAG affinity purification for identifying interacting proteins. Since Ku70 and Ku80 are two strong interactors of DNAPKc and participate in the NHEJ-mediated repair of damaged DNA, along with DNAPKc, we have also checked Ku70 and Ku80 as well.

The authors should show the proteins list identified in the mass spec analysis.

In our last submitted manuscript, we did show the mass spectrometry result of entire AF9-interacting protein through deposition in Mendeley repository and making it accessible through link

<https://data.mendeley.com/datasets/d2s63vgnmk/1>

This was also mentioned in the Reporting Summary that accompanied the last submitted manuscript. In this revised manuscript, along with the above link, we have also submitted our entire spectra file of mass spectrometry data in ProteomeXchange database that can be accessed through following credentials.

[Editorial Note: The username and password have been redacted to prevent the publication of confidential information]

Username: [REDACTED]

Password: [REDACTED]

Reviewer #3 (Remarks to the Author):

In this paper, the authors use a deletion analysis coupled with IP's to show that the AF9 YEATS domain is essential for the interaction with TFIID, but not with components of the SEC such as CDK9 and ELL. They identify several genes that have reduced expression (and reduced SEC and TBP binding) upon AF9 KD, and show that YEATS 43-60aa deletions fail to rescue gene expression. They go on to show that a Poly-Serine domain is essential for AF9 self association and interaction with TFIID. Interestingly, a p53 oligomerization domain restores self association and TFIID interaction, and the rest of the YEATS domain contributes to the TFIID interaction but not the self association function, as measured by IP's, RT-PCR and CHIP.

To analyse AF9 in a more dynamic system, the authors treat 293T cells with ionizing radiation and show that AF9 displays reduced oligomerization, reduced TFIID and SEC interactions, and overall the cells display reduced transcriptional activity, and this is associated with increased acetylation of AF9.

Acetylation of AF9 is controlled by PCAF on residue K339 (and deacetylated by HDAC5), and a K339Q acetyl mimic failed to rescue transcriptional defects in AF9 KD cells. With a series of detailed experiments they also show that DNA-PKc mediated phosphorylation at S395 reduces SEC binding to AF9, that a DNA-PKc interaction is dependent on K339 acetylation, and that components of NHEJ repair complexes are recruited to chromatin. The overall model is that acetylation of AF9 by PCAF reduces oligomerization and helps recruit DNA-PKc and disrupt SEC/TFIID binding to reduce transcription and enhance DNA repair upon irradiation.

This is an exciting, robust paper that builds on the author's previous work on AF9 function and significantly contributes to an important and expanding area of research. I have a few comments and questions, but overall this is excellent work.

We are ecstatic seeing such enthusiastic comments from this Reviewer regarding the quality of our work and its suitability for publication in *Nature Communications*. We have been happy to address all the comments made by this Reviewer as well.

Major questions:

1. Do the AF9 KD cells grow over the long term?

This is a very good point that the Reviewer has raised. In fact, we also had a similar question in our mind. In our experimental setup, AF9 knockdown (KD) cells do not survive beyond few splits. This is quite conceivable since AF9 knockdown majorly affects expression of proliferation-related genes within mammalian cells.

Do they have other defects other than reduced colony forming capabilities?

Apart from defect in colony formation, AF9 KD cells also show a defect in DNA repair capability even when cells are grown in normal cellular growth condition and thus show enhanced presence of γ -H2AX (Fig. 6N, compare lane 2 vs lane 3 in this revised version and Fig. S6G). This observation was also mentioned in our earlier submitted manuscript as well.

2. As the authors indicate, ENL has a similar structure to AF9, and the proteins are considered to be possibly functionally interchangeable. Is ENL expressed in 293T cells?

ENL does express within 293T cells as shown in Supplemental Figures S8D . In our experimental setup, the AF9 KD cells express the ENL protein as shown below. However, despite its presence, it seems that ENL can't complement the absence of AF9 for functional regulation. Thus, from our analyses, it seems that functions of both these proteins are not interchangeable.

Figure R9: Immunoblotting analysis showing overall effect of AF9 knockdown on expression of ENL protein within mammalian 293T cells.

In this revised manuscript, we have presented this data as Figure S8I.

If so, do ENL knockdowns impact expression of similar genes as the AF9 targets in Figure 1?

Yes, indeed, ENL KD cells also show similar expression defects as shown for AF9 as well (shown below)

Whether the AF9-containing SEC and ENL-containing SEC have separate functions in different steps of transcription cycle and thus needing the functions of both these proteins for expression of same genes is a question and hypothesis that needs to be tested and beyond the scope of this work. In this revised manuscript, this data has been presented as Fig. S8H.

3. Do ENL knockdowns also cause defects in colony formation of 293T cells?

Yes, indeed ENL KD cells also show colony formation defect as shown below.

In this revised manuscript, these data have been presented as Fig. S8F-G.

Minor points and questions:

1. Is there any evidence from the mass spec data for acetylation of AF9?

No, we have not performed any mass spectrometric analysis for identifying acetylation of AF9 protein. Our mass spectrometry analysis was designed for identifying only the interacting proteins.

2. In the paper, the authors consistently refer to AF9 as having a role in global transcription regulation. From Figure 1E there appear to be genes that are insensitive to loss of AF9. A few words explaining this discrepancy should be made in the discussion.

This is a very good suggestion. It could as well be that these sets of genes are not dependent on AF9 functions for their expression and thus are insensitive to the depletion of AF9. We have mentioned this aspect in the "Discussion" section of this revised version as suggested by this Reviewer.

3. Do other YEATS family (e.g. YEATS2, YEATS4) proteins contain a Poly-Ser stretch?

Based on the information from Uniprot, both the YEATS2 and YEATS4 do not possess the Poly-Ser domain as observed in AF9. Thus, their YEATS domain-mediated potential functional regulation could be different from that of AF9.

4. Although at one point the authors mention that all the work was done in 293T cells (except the final few experiments in HeLa cells), it would be helpful to indicate in the figure legends exactly what cells were used for each experiment.

This is very good suggestion and in this revised version, we have mentioned this point as suggested.

5. The legends should also include n numbers for all the experiments. Are the many blots representative of multiple or single experiments?

We are really sorry that we missed pointing this out in our last submitted manuscript. Most of our experiments are performed at least n=2 biological replicates and the presented data are representative of those biological replicates. To avoid repetition of this statement, we have mentioned this point in the methods section that describes immunoprecipitation assays.

6. The paper could overall use some editing for clarity.

We humbly accept the deficiency in our English writing capability. In this revised version, we have tried our level best to further edit in some places for making them clearer. However, if the paper is accepted, we would be more than happy to seek help from professional editing services.

References:

YADAV, D., GHOSH, K., BASU, S., ROEDER, R. G. & BISWAS, D. 2019. Multivalent Role of Human TFIID in Recruiting Elongation Components at the Promoter-Proximal Region for Transcriptional Control. *Cell Rep*, 26, 1303-1317 e7.

REVIEWERS' COMMENTS

Reviewer #1 (Remarks to the Author):

This revised manuscript has been improved in line with the comments raised and authors gave persuasive answers to the comments. I have no other concerns with the revised manuscript.

Reviewer #2 (Remarks to the Author):

The authors addressed most of my concerns and I think the manuscript is greatly improved. I recommend for acceptance the paper.

Reviewer #3 (Remarks to the Author):

I continue to be highly enthusiastic about this excellent piece of work. I find the result that ENL KDs reduce expression of key target genes but can't rescue AF9 KDs to be a very interesting new piece of data. Not enough work has been done on the possible separate functions of these two proteins. The reviewers have answered all of my questions and presented interesting new data, the paper is much improved and I am very supportive of it.

RE: NCOMMS-23-44203A

Post-translational modification-dependent oligomerization switch in regulation of global transcription and DNA damage repair during genotoxic stress

Prathama Talukdar, Sujay Pal, and Debabrata Biswas

Authors' Response to Reviewers' Comments:

We really appreciate all the efforts that the anonymous Reviewers have put into for improving our manuscript through their critical comments while reviewing our manuscript. Following is our response to Reviewer's comments:

Reviewer 1:

Comments: This revised manuscript has been improved in line with the comments raised and authors gave persuasive answers to the comments. I have no other concerns with the revised manuscript.

Response: We are happy that our response to this Reviewer's comments could satisfy the points that he/she had raised earlier. We also would like to thank the efforts put by this anonymous Reviewer for his/her efforts in improving our manuscript through the constructive comments that were made in the last revision.

Reviewer 2:

Comments: The authors addressed most of my concerns and I think the manuscript is greatly improved. I recommend for acceptance the paper.

Response: We are happy that the Reviewer has recommended the manuscript to be accepted for publication in Nature Communications. Like the other Reviewers, we also would like to thank the efforts put by this anonymous Reviewer for his/her efforts in improving our manuscript through the constructive comments that were made in the last revision.

Reviewer 3:

Comments: I continue to be highly enthusiastic about this excellent piece of work. I find the result that ENL KDs reduce expression of key target genes but can't rescue AF9 KDs to be a very interesting new piece of data. Not enough work has been done on the possible separate functions of these two proteins. The reviewers have answered all of my questions and presented interesting new data, the paper is much improved and I am very supportive of it.

Response: We are happy that like the last time, the Reviewer has shown extreme enthusiasm in our study for its publication in Nature Communications. Indeed, the data showing the relation between AF9 KD and ENL KD is extremely interesting and would be pursued in the lab as well. Further we also would like to thank the efforts put by this anonymous Reviewer for his/her efforts in improving our manuscript through the constructive comments and being constantly supportive of this study for its publication in Nature communications.